# FedGMKD: An Efficient Prototype Federated Learning Framework through Knowledge Distillation and Discrepancy-Aware Aggregation

**Jianqiao Zhang**
Department of Computer Science
Aberystwyth University
Aberystwyth, UK
jiz20@aber.ac.uk

**Caifeng Shan**[*]
School of Intelligence Science and Technology
Nanjing University
Nanjing, China
caifeng.shan@gmail.com

**Jungong Han**[*]
Department of Automation
Tsinghua University
Beijing, China
jungonghan77@gmail.com

## Abstract

Federated Learning (FL) faces significant challenges due to data heterogeneity across distributed clients. To address this, we propose FedGMKD, a novel framework that combines knowledge distillation and differential aggregation for efficient prototype-based personalized FL without the need for public datasets or server-side generative models. FedGMKD introduces Cluster Knowledge Fusion, utilizing Gaussian Mixture Models to generate prototype features and soft predictions on the client side, enabling effective knowledge distillation while preserving data privacy. Additionally, we implement a Discrepancy-Aware Aggregation Technique that weights client contributions based on data quality and quantity, enhancing the global model's generalization across diverse client distributions. Theoretical analysis confirms the convergence of FedGMKD. Extensive experiments on benchmark datasets, including SVHN, CIFAR-10, and CIFAR-100, demonstrate that FedGMKD outperforms state-of-the-art methods, significantly improving both local and global accuracy in Non-IID data settings.

## 1 Introduction

Federated Learning (FL) is a transformative approach to collaborative machine learning that allows multiple participants to train a shared model while maintaining data privacy by keeping datasets decentralized. This architecture mitigates privacy risks, particularly in sensitive domains such as healthcare and finance[3, 5, 6, 14, 23, 26]. FL enables clients to contribute to the global model without transferring raw data, reducing communication overhead while ensuring privacy. However, the Non-IID (non-independent and identically distributed) nature of client data poses challenges for model convergence and consistency. Heterogeneous data leads to divergence in model updates, slowing convergence and degrading performance across clients[14, 15, 23].

Personalized Federated Learning (pFL) has emerged as a solution to tailor models to individual client data distributions, improving local accuracy while maintaining global robustness[7, 8, 37].

---

[*]Joint corresponding authors.

38th Conference on Neural Information Processing Systems (NeurIPS 2024).

However, traditional pFL approaches relying on knowledge distillation (KD) often require public datasets, raising privacy concerns and complicating implementation[1, 9]. Straggler inefficiencies in traditional aggregation methods also delay convergence[25], highlighting the need for more effective pFL frameworks[29, 40].

To address these challenges, we propose FedGMKD, a novel framework that integrates Cluster Knowledge Fusion (CKF) and a Discrepancy-Aware Aggregation Technique (DAT) to enhance both local and global model performance. Our approach introduces several key innovations: first, by employing Gaussian Mixture Models (GMM), we generate prototype features and soft predictions for each class at the client side, which are aggregated at the server without the need for public datasets, thus preserving data privacy and effectively addressing the Non-IID problem. Second, DAT adjusts the aggregation process by weighting prototype features and soft predictions based on both the quality and the quantity, rather than the quantity, of client data, thus enhancing the global model's ability to generalize across diverse client data distributions. Furthermore, we conduct a rigorous theoretical analysis of the convergence and convergence rate of FedGMKD, providing strong guarantees for its effectiveness. Finally, experimental results on multiple benchmark datasets demonstrate that FedGMKD not only achieves state-of-the-art accuracy in local and global models but also significantly improves efficiency, making it highly effective in addressing the heterogeneity challenge while maintaining computational feasibility.

## 2 Related Work

### 2.1 Heterogeneity in Federated Learning

FL techniques have evolved significantly to address the challenges posed by the non-IID nature of data across diverse clients. Methods such as FedProx [22], Scaffold [15], and FedAlign [27] have substantially advanced the reduction of client-side bias by refining local updates to achieve better alignment with the global model. Nevertheless, these methods often rely on simplified assumptions about the underlying data distributions, which may prove inadequate in handling extreme variations in data heterogeneity, such as severe data imbalance and non-overlapping feature spaces. On the server side, solutions like FedOpt [32] and Agnostic Federated Learning [28] attempt to dynamically adjust aggregation strategies based on client contributions. However, despite these innovations, they continue to face significant challenges with scalability and often incur high computational costs, which limit their applicability in large-scale, highly diverse networks [20, 33, 38]. Moreover, approaches such as FedMix [39], which introduce synthetic data generation to approximate IID conditions, still struggle to accurately replicate the complex and highly variable data distributions encountered in real-world scenarios, particularly in non-IID environments.

### 2.2 Personalized Federated Learning

Personalization in federated learning has emerged as a crucial strategy to cater to diverse client-specific data characteristics. Strategies range from adapting global models for local refinement (e.g., FedPer [2] and Per-FedAvg [8]) to fully personalized approaches like MOCHA [34], pFedMe [35], and Ditto [21]. These personalized models often perform well on local data but can diverge significantly from the global model, leading to challenges in maintaining a cohesive learning strategy across the network. Additionally, the resource demands for training individualized models can be prohibitive, especially in scenarios with limited computational and bandwidth resources [7, 25].

### 2.3 Knowledge Distillation and Prototype Learning in FL

Knowledge distillation and prototype learning have made notable contributions to federated learning by enabling more efficient model training and facilitating knowledge transfer across distributed networks. KD-based approaches, such as FedMD [18] and FedDF [24], allow for the compression of model knowledge into more efficient representations, thereby reducing communication costs and enhancing scalability. However, these methods necessitate meticulous tuning of distillation parameters to achieve a balance between model complexity and performance [11]. FedProto [36] introduces prototype learning by aggregating class prototypes from clients to improve generalization across heterogeneous datasets. However, while FedProto improves local validation accuracy, it shows minimal improvement in global performance. As an extension of FedProto, FedHKD [4] integrates

knowledge distillation by introducing the concept of "hyper-knowledge", which involves sending the mean representations of both feature and soft predictions from clients to the server for global aggregation. While FedHKD significantly enhances client-side personalized model performance without relying on public datasets [4], it focuses mainly on improving the client models. It does not introduce substantial advancements in hyper-knowledge extraction techniques, nor does it achieve dual improvement for both client and server performance in theory, as knowledge distillation predominantly benefits the client-side models. Building upon FedHKD, our research addresses these limitations by not only improving the knowledge distillation process by prototype feature and soft predictions extraction but also benefiting the server-side models by using DAT. This ensures simultaneous improvement in both global model performance and client models.

## 3 Methodology

### 3.1 Problem Formulation

In the context of FL, we consider a scenario involving $n$ distinct clients, each holding a private dataset $\{\mathcal{D}_1, \mathcal{D}_2, \ldots, \mathcal{D}_n\}$, where each dataset reflects a potentially unique subset of the overall class distributions. This setup illustrates the fundamental Non-IID data challenge in FL, where the heterogeneity of each client's dataset $\mathcal{D}_i$ complicates the task of learning a generalized global model. Each client optimizes a local model $\mathbf{w}_i$ by minimizing a local loss function $F_i$, which is typically expressed as the empirical risk over its dataset $\mathcal{D}_i$:

$$F_i(\mathbf{w}_i) = \frac{1}{|\mathcal{D}_i|} \sum_{x \in \mathcal{D}_i} \ell(\mathbf{w}_i, x), \tag{1}$$

where $\ell$ denotes the loss function measuring the prediction error on an instance $x$.

To combine these local models into a global model, the FedAvg algorithm, introduced by McMahan et al. [26], applies a weighted averaging scheme based on the size of each client's dataset:

$$\mathbf{W}^r = \frac{1}{N} \sum_{i=1}^{n} |\mathcal{D}_i| \mathbf{w}_i^r, \tag{2}$$

where $\mathbf{W}^r$ represents the parameters of the global model after the $r$-th aggregation round, $\mathbf{w}_i^r$ denotes the parameters of the $i$-th client's model in the $r$-th round, and $N$ is the total number of samples across all clients, $N = \sum_{i=1}^{n} |\mathcal{D}_i|$. Our objective with FedGMKD is to tailor personalized models $\mathbf{w}_i$ for each client $i$ that not only achieve high accuracy on locally specific data but also enhance a global model $\mathbf{W}$ that excels across diverse client distributions. This task becomes particularly challenging in the presence of heterogeneous data, as naive aggregation approaches tend to diminish the effectiveness of the global model. **FedHKD** [4] has demonstrated significant success in addressing this issue by using hyper-knowledge distillation, which improves both global and personalized models through the exchange of mean representations and soft predictions. *Inspired by* this approach, **FedGMKD** enhances model performance on both the client and server sides by introducing Cluster Knowledge Fusion (CKF) based on Gaussian Mixture Models (GMM) for prototype feature and soft predictions generation and Discrepancy-Aware Aggregation Technique (DAT) based on data quality. This dual mechanism not only improves local model performance through better feature representation but also refines the global model by leveraging high-quality aggregated knowledge across clients, thereby effectively addressing the challenges posed by non-IID data distributions in federated learning.

### 3.2 Utilizing Cluster Knowledge Fusion

In Federated Learning (FL), traditional Knowledge Distillation (KD) techniques often rely on transferring knowledge from a complex central model (teacher) to simpler client models (students). This process typically requires a public dataset to generate soft labels representing the teacher's predictions in FL. Meanwhile, this approach raises significant privacy concerns and faces challenges, especially in Non-IID data environments, where the heterogeneity of client data exacerbates these issues, reducing the effectiveness of traditional KD methods in FL. To address these challenges and inspired by hyper-knowledge of FedHKD [4], we propose a Clustered Knowledge Fusion (CKF) approach based on Gaussian Mixture Models (GMM). CKF clusters client updates according to data

similarity, generating prototype features and soft predictions for each class, which are then aggregated at the server. These aggregated results form a synthetic dataset that serves as the foundation for knowledge distillation, enabling efficient knowledge transfer between the server and clients. Unlike traditional KD approaches, CKF not only eliminates the reliance on public datasets but also leverages client-specific representations, effectively addressing the Non-IID data problem and improving overall FL performance.

In the context of federated image classification, each client $i$ processes raw image data $x_i$ through a feature extraction function $F_{\phi_i}$, producing a feature representation $h_i$, and a classifier function $C_{\omega_i}$ which maps $h_i$ into a soft prediction vector $z_i$:

$$h_i = F_{\theta_i}(x_i), \quad z_i = \text{Softmax}\left(C_{\psi_i}(h_i)\right), \tag{3}$$

where $F_{\theta_i}(\cdot)$ and $C_{\psi_i}(\cdot)$ represent the feature extractor and classification function for client $i$, respectively. The feature vector $h_i$ encodes the latent features of the input data, and $z_i$ represents the soft prediction vector, indicating the probability distribution over the classes.

When the features and soft predictions have been extracted, GMM are employed to cluster client updates based on data similarity prior to aggregation.

The responsibility $\gamma_m(\mathbf{x}_i^j)$ indicates the probability that the data point $\mathbf{x}_i^j$ (which can be a feature vector or a soft prediction vector) belongs to the $m$-th Gaussian component. It is calculated as:

$$\gamma_m(\mathbf{x}_i^j) = \frac{\pi_m \cdot \mathcal{N}(\mathbf{x}_i^j; {}^-{}_m, \mathbf{\Sigma}_m)}{\sum_{s=1}^{M} \pi_s \cdot \mathcal{N}(\mathbf{x}_i^j; {}^-{}_s, \mathbf{\Sigma}_s)}, \tag{4}$$

Here, $\pi_m$ represents the mixture coefficient of the $m$-th Gaussian component, where $\sum_{m=1}^{M} \pi_m = 1$. The term $\mathcal{N}(\mathbf{x}_i^j; {}^-{}_m, \mathbf{\Sigma}_m)$ denotes the Gaussian probability density function evaluated at $\mathbf{x}_i^j$, with ${}^-{}_m$ being the mean vector and $\mathbf{\Sigma}_m$ being the covariance matrix of the $m$-th Gaussian component. The denominator sums over all $M$ Gaussian components, normalizing the responsibility values so that they sum to 1 across all components. These responsibility values indicate the contribution of each Gaussian component to the data point $\mathbf{x}_i^j$, and they are later used to compute the prototype features and soft predictions by weighting the means and predictions of each Gaussian component. Using the responsibility values, the prototype features and soft predictions for class $j$ in the local dataset of client $i$ are calculated as follows:

$$\hat{h}_i^j = \sum_{m=1}^{M} \gamma_m(\mathbf{h}_i^j){}^-{}_{m_j}, \quad \hat{q}_i^j = \sum_{m=1}^{M} \gamma_m(\mathbf{z}_i^j)\mathbf{z}_{m_j}, \tag{5}$$

where $\hat{h}_i^j$ denotes the prototype feature for class $j$ at client $i$, synthesized from the cluster knowledge, and $\hat{q}_i^j$ represents the prototype soft prediction for class $j$ at client $i$, calculated based on the responsibility values. Here, ${}^-{}_{m_j}$ represents the mean vector of the $m$-th Gaussian component for class $j$, and $\mathbf{z}_{m_j}$ represents the soft prediction vector corresponding to the $m$-th Gaussian component.

CKF is derived by integrating these GMM-based prototype features and their corresponding prototype soft predictions. For class $j$ in the local dataset of client $i$, CKF is defined by combining the prototype features and their prototype soft prediction:

$$K_i^j = (\hat{h}_i^j, \hat{q}_i^j). \tag{6}$$

If there are $j$ classes, then the full CKF of client $i$ is:

$$K_i = \bigcup_{j=1}^{J} \left(\hat{h}_i^j, \hat{q}_i^j\right) \tag{7}$$

A flow diagram illustrating the computation of CKF is provided in Supplementary Section A.1.

## 3.3 Discrepancy-Aware Aggregation Technique

The aggregation phase in FedGMKD amalgamates CKF from each client to construct a global representation of CKF for a given class $j$ at each iteration $r + 1$. For the given class $j$, the global CKF represented by the aggregated prototype features $\mathbf{H}_j^{r+1}$ and soft predictions $\mathbf{Q}_j^{r+1}$. These are calculated as follows:

$$\mathbf{H}_j^{r+1} = \sum_{i=1}^{n} w_i' \cdot \hat{h}_i^{j,r}, \quad \mathbf{Q}_j^{r+1} = \sum_{i=1}^{n} w_i' \cdot \hat{q}_i^{j,r}, \tag{8}$$

where $w_i'$ is the weight corresponding to client $i$'s contribution, reflecting both the volume and the quality of the data contributed by client $i$.

In traditional FL approaches such as FedAvg, the server aggregates models by averaging the parameters submitted by each client. This method assumes that data across different clients are identically and independently distributed (IID). However, tin real-world applications, this assumption often fails due to Non-IID data distributions, leading to sub-optimal global models when client data varies significantly in distribution and relevance. Most existing methods primarily weight client contributions based on data volume, neglecting the quality of the data, which can further exacerbate this issue.

To address these challenges, FedGMKD introduces a Discrepancy-Aware Aggregation method that evaluates both the volume and the quality of data each client contributes. This is achieved by quantifying how well the local data aligns with the global data distribution. The aggregation process is refined by incorporating a measure of the discrepancy between local model prototype predictions and the aggregated global prototype predictions, thereby enhancing the robustness and accuracy of the federated model.

To begin, the initial weights for each client and class are calculated based on the proportion of samples for that class across all clients:

$$w_{i,j}^{\text{init}} = \frac{N_i^j}{\sum_{i=1}^{n} N_i^j}, \tag{9}$$

where $N_i^j$ is the number of samples of class $j$ in client $i$'s local dataset. This ensures that the initial weight $w_{i,j}^{\text{init}}$ reflects the proportion of class $j$ samples that client $i$ contributes to the global dataset for that class.

After this, the final aggregation weights are adjusted based on the discrepancies between the local and global data distributions. This discrepancy is quantified using the Kullback-Leibler (KL) divergence and is balanced by the initial weight $w_{i,j}^{\text{init}}$ for each category $j$:

$$w_i' = \frac{\text{ReLU}\left(w_{i,j}^{\text{init}} - a \cdot d_i^j + b\right)}{\sum_{i=1}^{n} \text{ReLU}\left(w_{i,j}^{\text{init}} - a \cdot d_i^j + b\right)}, \tag{10}$$

where $w_{i,j}^{\text{init}}$ is the initial weight for class $j$ on client $i$, reflecting the proportion of class $j$ samples contributed by client $i$ to the global dataset. $d_i^j$ represents the KL divergence between the client and server distributions for class $j$, and $a$ and $b$ are adjustment parameters that control the sensitivity of the weight updates based on the discrepancies.

The KL divergence $d_i^j$ between the client $i$'s local distribution and the server's global distribution for class $j$ is calculated as follows:

$$d_i^j = D_{\text{KL}}\left(\hat{q}_i^j \parallel \hat{Q}_j\right) = \hat{q}_i^j \log \frac{\hat{q}_i^j}{\hat{Q}_j}, \tag{11}$$

where $\hat{q}_i^j$ and $\hat{Q}_j$ are the predicted probabilities for class $j$ in the local client $i$ and the global server distributions, respectively. This KL divergence measures the discrepancy between the local and

global predictions specifically for class $j$, allowing the model to adjust the aggregation weights based on this class-specific difference.

For each class $j$, the global CKF is computed by aggregating the local CKF from all clients using the discrepancy-aware weights $w_i'$. These global CKF values, represented by $\mathbf{H}_j^{r+1}$ and $\mathbf{Q}_j^{r+1}$, encapsulate the collective knowledge from all clients for class $j$. After calculating the global CKF for each class $j$, the complete global CKF, denoted as $G^{r+1}$, is constructed by taking the union of the global CKF for all classes:

$$G^{r+1} = \bigcup_{j=1}^{j} \left( \mathbf{H}_j^{r+1}, \mathbf{Q}_j^{r+1} \right), \tag{12}$$

This complete global CKF $G^{r+1}$ serves as the updated global knowledge representation, which is used to guide future iterations of model updates. A flow diagram illustrating the computation of DAT is provided in Supplementary Section A.2.

### 3.4 Local Training Objective Function

After the server completes the aggregation, the updated global CKF $G^{r+1}$ is sent to the clients selected for the next FL round to aid in their local training. For client $i$, with dataset $\mathcal{D}_i$, the local training objective integrates the empirical risk with regularization terms designed to align the local model with the global CKF. The loss function for client $i$ is defined as:

$$L(\mathcal{D}_i, \mathbf{w}_i) = \frac{1}{|\mathcal{D}_i|} \sum_{(x_k, y_k) \in \mathcal{D}_i} \ell \left( C_{\psi_i} \left( F_{\theta_i}(x_k) \right), y_k \right)$$
$$+ \lambda \frac{1}{|\mathcal{D}_i|} \sum_{(x_k, y_k) \in \mathcal{D}_i} \left\| F_{\theta_i}(x_k) - \mathbf{H}_{y_k}^{r+1} \right\|_2^2 + \frac{\gamma}{n} \sum_{j=1}^{n} \left\| \frac{C_{\psi_i} \left( \mathbf{H}_j^{r+1} \right)}{T} - \frac{\mathbf{Q}_j^{r+1}}{T} \right\|_2^2. \tag{13}$$

where $|\mathcal{D}_i|$ denotes the number of samples in the dataset owned by client $i$, $\ell(\mathcal{C}_{\psi_i}, y_k)$ denotes the cross-entropy loss function, $\|\cdot\|_2^2$ denotes the Euclidean norm, and $\lambda$ and $\gamma$ are hyper-parameters. Note that $\mathbf{H}_{y_k}^{r+1}$ represents the global prototype feature for class $j$, and $\mathbf{Q}_j^{r+1}$ denotes the global soft predictions for class $j$ at iteration $r + 1$. The term $C_{\psi_i}(\mathbf{H}_j^{r+1})$ represents the local classifier's predictions on the global prototype feature $\mathbf{H}_j^{r+1}$ and T is the temperature of KD.

The loss function consists of three terms: the empirical risk formed using predictions and ground-truth labels, and two regularization terms that utilize the global CKF. The first term is the empirical risk, represented by the cross-entropy loss function. This term encourages the local model to perform well on its own data. The second term, known as the feature alignment loss, aligns the local feature extractor with the global CKF by minimizing the squared Euclidean distance between the local feature representations $F_{\theta_i}(x_k)$ and the globally aggregated CKF features $\mathbf{H}_{y_k}^{r+1}$. This regularization encourages the local feature extractor to produce similar feature representations to the global prototype features for each corresponding class, improving consistency between the local and global models. The third term, called the knowledge alignment loss, ensures predictive consistency across federated learning by minimizing the discrepancy between the local classifier's predictions on the global prototype features and the global soft predictions for each class $j$. Specifically, by using Euclidean distance, which is non-negative and convex, these terms effectively regularize the local models to be more consistent with the global CKF.

### 3.5 Overall Framework of FedGMKD

FedGMKD integrates CKF and DAT to enhance both local and global model performance. The framework operates iteratively, as outlined in Algorithm 1. The server initializes the global model $\mathbf{W}^0 = (F^0, C^0)$, where $F^0$ and $C^0$ denote the parameters of the global feature extractor and classifier, respectively. The global CKF $G^0$ is also initialized, comprising global prototype features and soft predictions for each class. At each global epoch, the server transmits the global model $\mathbf{W}^{r-1}$ and

global CKF $G^{r-1}$ to the selected clients. Clients update their local models by minimizing a composite loss function comprising three components: (1) the empirical risk, representing the cross-entropy between predicted and ground truth labels; (2) the feature alignment loss, measured by the Euclidean distance between local and global prototype features; and (3) the knowledge alignment loss, measured by the Euclidean distance between local classifier outputs on global prototype features and the global soft predictions. Upon completing local training, clients compute local CKF and transmit these, along with their updated models, back to the server. The server aggregates the received CKF using DAT, which adjusts client contributions based on data volume and quality. The global CKF and model are updated accordingly. This process repeats across multiple federated learning rounds.

---

**Algorithm 1** FedGMKD

---

**Require:** Distributed datasets across $n$ clients, $D = \{D_1, D_2, \ldots, D_n\}$; client participation rate $\mu$; hyper-parameters $\lambda$ and $\gamma$; temperature $T$; number of global epochs $R_r$.
**Ensure:** Updated global model $\mathbf{W}^{R_r+1}$ and personalized local models $\{\mathbf{w}_1^{R_r+1}, \mathbf{w}_2^{R_r+1}, \ldots, \mathbf{w}_i^{R_r+1}\}$.
 1: Server initializes the global model $\mathbf{W}^0$ and global CKF $G^0$ for each class.
 2: **for** $r = 1$ to $R_r$ **do**
 3:  Server selects $i$ clients for participation.
 4:  Server broadcasts $\mathbf{W}^{r-1}$ and $G^{r-1}$ to the selected clients.
 5:  **for** each selected client $i$ **do**
 6:    Client $i$ initializes local model $\mathbf{w}_i^{r-1}$ from $\mathbf{W}^{r-1}$.
 7:    **if** $r == 1$ **then**
 8:      Client $i$ updates $\mathbf{w}_i^r$ using Equation 1.
 9:      Client $i$ computes initial CKF $K_i^r$ using Equations 3, 4, 5, 6, and 7.
10:    **else**
11:      Client $i$ updates $\mathbf{w}_i^r$ using Equation 13, incorporating $G^{r-1}$.
12:      Client $i$ computes CKF $K_i^r$ using Equations 3, 4, 5, 6, and 7, and computes divergence $d_i^r$ for each class using Equation 11.
13:    **end if**
14:    Client $i$ sends $\mathbf{w}_i^r$, $K_i^r$, and $d_i^r$ (if $r > 1$) back to the server.
15:  **end for**
16:  **if** $r == 1$ **then**
17:    Server averages CKFs and models from clients to update global CKF $G^{r+1}$ and global model $\mathbf{W}^{r+1}$ using Equations 8, 9 and 12.
18:  **else**
19:    Server computes weights for each class of each client using Equation 10.
20:    Server updates global CKF $G^{r+1}$ using Equations 8 and 12 based on the computed weights.
21:    Server updates global model $\mathbf{W}^{r+1}$ using weighted averaging of the models.
22:  **end if**
23:  Server sends $\mathbf{W}^{r+1}$, $G^{r+1}$ (if $r > 1$) back to client $i$.
24: **end for**
25: **return** Updated global model $\mathbf{W}^{R_r+1}$ and local models $\{\mathbf{w}_1^{R_r+1}, \mathbf{w}_2^{R_r+1}, \ldots, \mathbf{w}_i^{R_r+1}\}$.

---

### 3.6 Convergence Analysis

Given the Non-IID nature of data across clients in FedGMKD, we establish two theorems to describe the framework's convergence under well-defined mathematical assumptions.

## Theorem 1: FedGMKD Convergence

Under Assumptions 1-5 (A.6.1), for any client $i$, after $R$ global communication rounds, the expected global loss function is bounded as:

$$\frac{1}{R} \sum_{r=1}^{R} \sum_{i=1}^{n} w_i' \mathbb{E}\left[\|\nabla F_i(\mathbf{w}_i^r)\|^2\right] \leq \frac{F(\mathbf{W}^1) - F^*}{\eta R^2} + \sigma^2 + \frac{L\eta R G^2}{2}, \tag{14}$$

The detailed proof is provided in A.6.3.

## Theorem 2: FedGMKD Convergence Rate

Under Assumptions 1-5 (A.6.1), for any client $i$, after $R$ global communication rounds, the convergence rate of the global loss function $F(\mathbf{W})$ is bounded as follows:

$$F(\mathbf{W}^R) - F^* \leq \frac{C_1}{R} + C_2, \tag{15}$$

where $F(\mathbf{W})$ is the global loss function, $F^*$ represents the lower bound of $F(\mathbf{W})$, and $C_1$ and $C_2$ are constants that depend on variance $\sigma^2$, Lipschitz constant $L$, learning rate $\eta$, and the number of local steps.

The detailed proof is provided in A.6.3.

## 4 Experiments

### 4.1 Datasets

We evaluate FedGMKD on three widely used FL benchmark datasets, selected to cover a range of task complexities, demonstrating the method's robustness and scalability.

**SVHN** [30]: The Street View House Numbers (SVHN) dataset consists of over 600,000 labeled digit images from street scenes. It presents varied imaging conditions, such as lighting and background differences, making it useful for testing under non-IID scenarios. Despite these variations, SVHN is considered relatively simple due to its large dataset size and clear digit images.

**CIFAR-10** [17]: CIFAR-10 comprises 60,000 32x32 color images across 10 classes, with 6,000 images per class. It is a standard benchmark for image classification in both centralized and federated learning, providing moderate complexity due to the diverse nature of the images.

**CIFAR-100** [17]: Similar to CIFAR-10 but with 100 classes, CIFAR-100 contains 60,000 images with 600 images per class. It poses a more challenging task, given the finer granularity and increased class variability, making it especially difficult in federated learning with non-IID data.

### 4.2 Models

For our experiments, we adopt the ResNet18 architecture [10], which has consistently demonstrated superior performance across a wide range of learning tasks, outperforming traditional Convolutional Neural Networks (CNNs). ResNet18 incorporates residual connections to effectively address the vanishing gradient issue, enabling the training of deeper networks while maintaining computational efficiency.

### 4.3 Baselines

To establish a comprehensive comparison, our evaluation includes a diverse set of baselines, encompassing both well-established methods and recent advances in federated learning. These baselines include: **FedAvg** [26], the foundational algorithm in federated learning; **FedProx** [22], addressing data heterogeneity; **MOON** [19], focusing on model personalization; **FedMD** [18] and **FedGen** [40], which utilize public datasets and generative models, respectively, to enhance performance under Non-IID conditions; **FedProto** [36] and **PFL** [13], which employ prototype and clustering learning methods to handle data disparities in federated learning; and **FjORD** [12], which introduces an ordered dropout mechanism to enable fair and accurate federated learning under heterogeneous target distributions.

### 4.4 Experimental Setting

The models were implemented and run using PyTorch [31] with 2 NVIDIA A100 GPUs. The Adam optimizer [16] was used for model training in all experiments. The learning rate was initialized to 0.001 and decreased every 10 iterations with a decay factor of 0.5, while the hyper-parameter in

Adam was set to 0.5. The number of global communication rounds was set to 50, and the number of local epochs was set to 3. The size of a data batch was set to 64, and the participating rate of clients was set to 1. For all datasets (SVHN [30], CIFAR-10, and CIFAR-100 [17]), the latent dimension of data representation was set to 32.

**Hyper-parameters**: For the FedProx [22] algorithm, the hyper-parameter $\mu_{prox}$ was set to 0.5. For the MOON [19] algorithm, the proximal term's hyper-parameter $\mu_{moon}$ was set to 1. In FedGen [40], a Multi-Layer Perception (MLP)-based architecture with a hidden dimension of 512 was employed for the generative model. Latent, noise, and input/output dimensions were tailored to each dataset, and the generative model was trained for 5 epochs per global round. The ratio of the generative batch-size to the training batch-size was 0.5 (generative batch-size set to 32). Parameters $\alpha_{generative}$ and $\beta_{generative}$ were initialized at 10 with a decay factor of 0.98 per global round. FedMD [18] used a regularization hyper-parameter $\lambda_{md}$ of 0.05, and the public dataset size matched the clients' local training dataset size. FedProto [36] had a regularization parameter $\lambda_{proto}$ set to 0.05. For FPL [13], the regularization parameter $\lambda_{FPL}$ was set to 0.1, with the number of prototypes per class fixed at 10. For FjORD [12], the unique dropout rate $\delta$ was set to 0.5, and the temperature for knowledge distillation $T_{fjord}$ was set to 0.7, following the original paper's settings. Finally, in our proposed FedGMKD, hyper-parameters $\lambda$ and $\gamma$ were set to 0.6. The clustering centers were dynamically adjusted between 2-7 based on the data distribution in each category, with the parameters $a$ and $b$ for differential aggregation both set to 0.2, and the temperature $T$ for knowledge distillation was set to 0.6.

## 4.5 Results

Table 1: Results on data partitions generated from Dirichlet distribution with the concentration parameter $\beta = 0.5$. The number of clients is 10, 20, and 50; the clients utilize 10%, 20%, and 50% of the datasets. A single client's averaged wall-clock time per round is measured across 2 A100 GPUs in a parallel manner. The reported local and global accuracies are the averages of the last 5 rounds.

| Dataset | Scheme | Local Acc | | | Global Acc | | | Avg Time (S) | Pub Data |
|---|---|---|---|---|---|---|---|---|---|
| | | 10 | 20 | 50 | 10 | 20 | 50 | | |
| SVHN | FedAvg | 84.29 | 85.20 | 85.67 | 81.98 | 87.32 | 89.72 | 168.44 | No |
| | FedProx | 85.25 | 86.38 | 86.08 | 81.71 | 87.40 | 88.74 | 229.17 | No |
| | Moon | 84.11 | 85.43 | 85.43 | 81.95 | 86.90 | 88.97 | 358.14 | No |
| | FedGen | 85.18 | 85.10 | 84.96 | 81.96 | 86.02 | 88.52 | 205.37 | No |
| | FedMD | 85.45 | 85.90 | 86.31 | 82.04 | 87.30 | 89.91 | 611.33 | Yes |
| | FedProto | 85.58 | 86.44 | 86.85 | 81.34 | 86.97 | 89.79 | 346.13 | No |
| | FPL | 85.37 | 86.02 | 85.87 | 79.81 | 85.64 | 88.76 | 522.83 | No |
| | FjORD | 85.13 | 85.97 | 86.21 | 81.56 | 85.09 | 89.36 | 380.74 | No |
| | **FedGMKD** | **86.26** | **87.43** | **87.16** | **82.64** | **87.78** | **90.17** | 312.52 | No |
| CIFAR10 | FedAvg | 55.75 | 58.76 | 61.51 | 46.62 | 52.61 | 51.53 | 98.94 | No |
| | FedProx | 57.46 | 58.91 | 62.94 | 47.97 | 53.13 | 56.04 | 126.56 | No |
| | Moon | 58.61 | 59.12 | 62.42 | 46.89 | 50.16 | 57.29 | 221.19 | No |
| | FedGen | 59.46 | 60.17 | 61.03 | 48.09 | 51.55 | 52.62 | 122.35 | No |
| | FedMD | 60.15 | 62.05 | 63.37 | 48.32 | 53.73 | 57.69 | 410.19 | Yes |
| | FedProto | 59.77 | 62.85 | 64.98 | 48.97 | 50.88 | 57.12 | 229.40 | No |
| | FPL | 60.95 | 62.74 | 64.49 | 47.19 | 52.04 | 58.35 | 295.97 | No |
| | FjORD | 59.62 | 63.36 | 63.61 | 49.18 | 53.22 | 58.74 | 252.34 | No |
| | **FedGMKD** | **61.78** | **64.04** | **65.69** | **49.78** | **55.16** | **60.31** | 251.55 | No |
| CIFAR100 | FedAvg | 15.39 | 17.10 | 21.09 | 14.51 | 18.98 | 22.21 | 97.02 | No |
| | FedProx | 16.45 | 17.56 | 21.91 | 16.06 | 19.67 | 23.35 | 120.36 | No |
| | Moon | 15.46 | 18.03 | 21.25 | 15.19 | 18.16 | 21.37 | 201.91 | No |
| | FedGen | 14.08 | 17.05 | 19.54 | 14.88 | 19.05 | 23.16 | 148.58 | No |
| | FedMD | 13.25 | 19.03 | 21.93 | 15.96 | 19.20 | 23.75 | 482.76 | Yes |
| | FedProto | 15.70 | 18.63 | 22.50 | 15.38 | 17.13 | 18.72 | 206.12 | No |
| | FPL | 15.93 | 18.24 | 21.96 | 15.37 | 18.19 | 21.59 | 373.09 | No |
| | FjORD | 15.94 | 19.91 | 22.60 | 16.93 | 21.45 | 22.86 | 226.73 | No |
| | **FedGMKD** | **17.16** | **20.96** | **23.57** | **16.97** | **21.56** | **24.63** | 275.60 | No |

The proposed method, FedGMKD, consistently demonstrates superior performance across the benchmark datasets, both in terms of local and global accuracy, often outperforming other federated learning methods. On the SVHN dataset, FedGMKD significantly improves both local and global test accuracies over FedAvg across different client counts. Compared to FedProto, FedGMKD exhibits notable improvements, with local accuracy gains ranging from 0.31% to 0.99%, and global accuracy improvements ranging from 0.38% to 1.3%. When compared to FPL, FedGMKD demonstrates even more substantial improvements, with local accuracy gains of 0.89% to 1.41%, and global accuracy gains of 1.41% to 2.83%. FjORD performs competitively in this dataset, showing results second only to FedGMKD, particularly in global accuracy. However, FedGMKD still outperforms FjORD by up to 1.46% in local accuracy and 2.69% in global accuracy, demonstrating its advantage in Non-IID environments.

On CIFAR-10, FedGMKD achieves substantial improvements over FedAvg, with local accuracy increases of 4.18% to 6.03% and global accuracy gains of 2.55% to 8.78%. FedGMKD also surpasses FedProto and FPL, showing gains of up to 2.01% in local accuracy and 4.28% in global accuracy compared to FedProto, and up to 1.3% in local accuracy and 3.12% in global accuracy compared to FPL. In this case, while FjORD also performs well, FedGMKD remains superior, particularly in terms of global accuracy, further showcasing the robustness of the proposed approach in handling Non-IID data. Meanwhile, on CIFAR-100, FedGMKD achieves significant gains over FedAvg, FedProx, and FPL in both local and global accuracies across all client counts. FedGMKD shows local accuracy improvements of 1.23% to 2.72% compared to FPL, and global accuracy gains of 1.6% to 3.37%. FjORD's performance in CIFAR-100, while strong in some settings, is still notably behind FedGMKD, particularly as the number of clients increases, where FedGMKD's clustering-based approach becomes increasingly effective.

In terms of computational efficiency, FedGMKD incurs a modest increase in training time compared to FedAvg but remains competitive given its substantial accuracy improvements. Compared to FedProx, FedGMKD shows comparable or slightly better training efficiency while maintaining superior accuracy. FedGMKD is also significantly more efficient than FPL, achieving better accuracy with reduced computational overhead. FjORD shows efficiency similar to FedProto but requires more time than FedGMKD on SVHN and CIFAR10. Overall, FedGMKD strikes an optimal balance between accuracy and computational requirements. The additional computational burden of FedGMKD is justified by the substantial gains in local and global accuracy through CKF and DAT. These improvements suggest that FedGMKD offers a highly effective and efficient solution for real-world federated learning scenarios, particularly in the presence of Non-IID data distributions.

## 5  Conclusion

This paper presents FedGMKD, a novel federated learning framework that addresses data heterogeneity without requiring public datasets and complex server-side models. Through Cluster Knowledge Fusion (CKF) and Discrepancy-Aware Aggregation Technique (DAT), FedGMKD achieves superior local and global accuracy across various benchmark datasets, outperforming methods like FedAvg, FedProto, and FPL. Meanwhile, theoretical convergence guarantees and experimental results validate its effectiveness. While FedGMKD introduces moderate computational overhead, the accuracy gains justify this cost. Future work will focus on improving computational efficiency and scalability to better handle large, complex datasets, enhancing its applicability to broader federated learning scenarios.

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

# A    Supplementary

## A.1    Flow Diagram Demonstrating the Computation of CKF

The flow diagram in Figure 1 illustrates the process of computing Cluster Knowledge Fusion (CKF) within the Federated Learning (FL) framework. This process integrates feature extraction, Gaussian Mixture Model (GMM) clustering, and the synthesis of prototype features and soft predictions from clients. Following this, aggregation occurs at the server using the Discrepancy-Aware Aggregation Technique (DAT), facilitating the alignment of global knowledge with local client distributions.

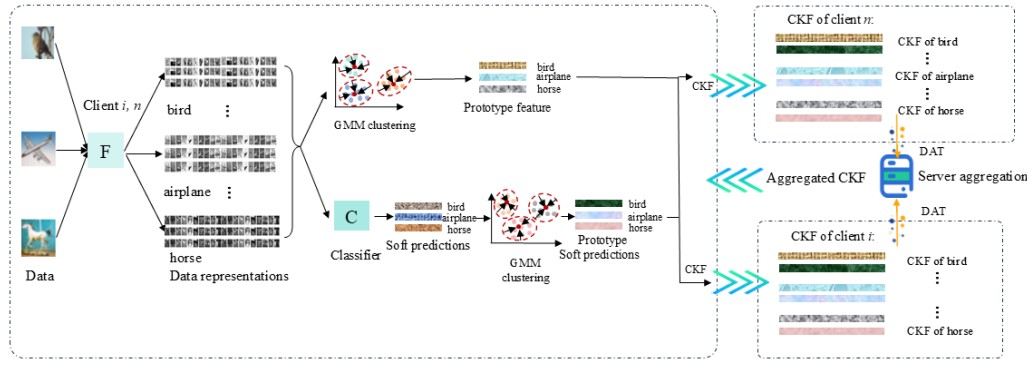

Figure 1: Flow diagram demonstrating the computation of Cluster Knowledge Fusion (CKF) in Federated Learning. The diagram highlights the steps involved in extracting features, generating soft predictions, and performing GMM clustering to compute prototype features and predictions, followed by the aggregation of CKF at the server.

As shown in Figure 1, the computation of CKF begins at each participating client after local training. Each client, $i$, first fine-tunes its local model consisting of a feature extractor $F_{\theta_i}(\cdot)$ and a classifier $C_{\psi_i}(\cdot)$. The CKF computation for class $j$ occurs in three major steps: the feature extractor processes raw input data to generate latent data representations for each class (e.g., bird, airplane, horse). These data representations capture the essential features of the input data at the client level. Using the generated data representations, GMM clustering is applied to compute the prototype features for each class. Concurrently, the classifier generates soft predictions for these data representations, which are also clustered using GMM to derive prototype soft predictions for each class. After obtaining the prototype features and soft predictions for each class, they are aggregated to form the CKF. The CKF encapsulates the most representative features and soft predictions for each class, thereby summarizing the local client knowledge in a form that is robust and suitable for global aggregation at the server. In this way, CKF computation reduces the redundancy inherent in the client data while maintaining critical data characteristics, ensuring that local model updates are aligned with class-specific feature distributions. The CKF values are then transmitted to the server, where the DAT is employed to merge the CKF across clients, resulting in a robust global CKF that reflects the diverse data distributions across the FL network.

## A.2    Flow Diagram Demonstrating the Computation of DAT

The flow diagram in Figure 2 illustrates the process of computing the Discrepancy-Aware Aggregation Technique (DAT) within the Federated Learning (FL) framework. This technique enhances the aggregation of Cluster Knowledge Fusion (CKF) from clients by incorporating both data quantity and quality, ensuring that the global CKF is a robust representation aligned with diverse local client distributions.

As shown in Figure 2, the computation of DAT begins by calculating an initial weight for each client's CKF. Each client's initial weight $w_{i,j}^{\text{init}}$ for class $j$ is derived based on the proportion of samples for that class relative to the total samples across all clients, reflecting each client's contribution in terms of data volume. This initial aggregation step combines CKF from all clients to generate preliminary

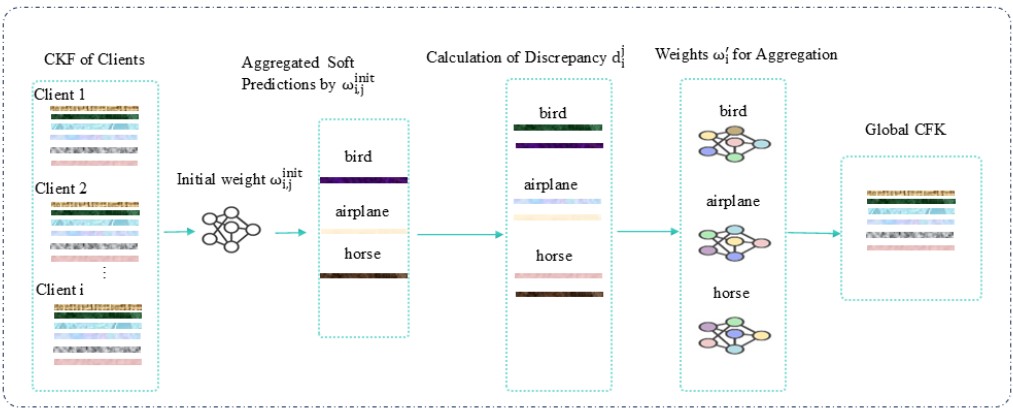

Figure 2: Flow diagram demonstrating the computation of Discrepancy-Aware Aggregation Technique (DAT) in Federated Learning. The diagram details the steps involved in computing initial weights, aggregating soft predictions, calculating discrepancies, and performing the final aggregation of CKF at the server.

aggregated soft predictions. Following this, the server evaluates the alignment between each client's local data distribution and the preliminary global soft predictions by calculating the Kullback-Leibler (KL) divergence $d_i^j$ for each client $i$ and class $j$. This discrepancy measure serves to quantify the quality of each client's CKF, indicating how closely each client's data aligns with the global distribution. Based on the calculated discrepancies, the server adjusts the initial weights, producing the final aggregation weights $w_i'$ for each client. These final weights balance the data volume and data quality, allowing clients with more aligned distributions to have a greater influence in the aggregation process. Finally, the server aggregates the CKFs from all clients using the discrepancy-aware weights $w_i'$, resulting in a robust global CKF for each class. This global CKF incorporates the essential knowledge from all clients while accounting for variations in data quality, effectively addressing the challenges posed by Non-IID data distributions in FL.

## A.3 Visualization of FedGMKD

The flow diagram in Figure 3 illustrates the iterative process of CKF computation and model aggregation DAT within the FedGMKD framework. This visualization helps clarify the operational phases of training local models, extracting CKF, and DAT aggregating updates at the server, leading to a refined global model.

In Figure 3, the FedGMKD process is visualized as follows: During Global Round 1, clients (e.g., Client $i$ and Client $n$) train their local models using their private data and extract CKF from these models. The local CKF and model updates are then sent to the server. Upon receiving updates from all participating clients, the server employs the DAT to aggregate the local CKF and model updates, refining both the global CKF and the global model.

In subsequent rounds (e.g., Global Round 2 through Global Round $T$), the server broadcasts the updated global CKF and global model to the clients. Each client integrates the global CKF into its local training process, updating its model based on both the global CKF and its own local data. After training, the clients extract new CKF values from their updated models and send these back to the server for further DAT aggregation. This iterative process continues across multiple global rounds, progressively refining both the global model and CKF to account for the diversity and heterogeneity of client data, thereby improving the personalization and generalization of the federated learning model.

By leveraging CKF at each iteration, the FedGMKD framework ensures that the global model retains a balanced understanding of the local client data distributions while adapting to the variations in the feature and label spaces across different clients. This process addresses the challenges posed by non-IID data, leading to a more robust and well-personalized federated learning system.

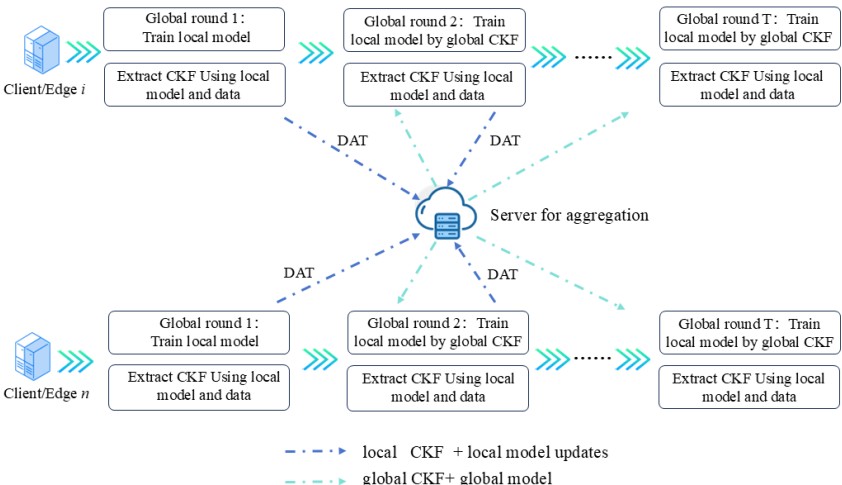

Figure 3: Visualization of the FedGMKD framework. Each client trains a local model and extracts CKF using its local data. The server aggregates the CKF and model updates using Discrepancy-Aware Aggregation Technique (DAT) to improve the global CKF and model. This process iterates over multiple global rounds.

## A.4 Ablation Study

### A.4.1 Handling Data Heterogeneity

To evaluate the robustness of FedGMKD in addressing data heterogeneity, we conducted experiments on the CIFAR-10 and SVHN datasets under varying degrees of Non-IID conditions, controlled by the Dirichlet distribution parameter $\beta$. A smaller $\beta$ indicates greater heterogeneity in the data distribution, with clients typically possessing only one or two classes in their local datasets; a larger $\beta$ suggests a more homogeneous distribution of data across clients. Specifically, when $\beta = 0.2$, the local datasets exhibit high heterogeneity, whereas $\beta = 5$ produces nearly homogeneous datasets.

For these experiments, we utilized datasets from Torchvision, where the global test dataset was directly obtained by loading the standard test sets for CIFAR-10 and SVHN without any sampling. To create the local datasets, we partitioned each dataset into $n$ clients (where $n$ denotes the number of clients) using a Dirichlet distribution with parameter $\beta$. Each client's dataset was subsequently divided into a local training set and a local test set in a 75%/25% split. The results of these experiments, which compare the performance of various methods under $\beta = 0.2$ and $\beta = 5$ settings for both CIFAR-10 and SVHN, are summarized in Table 2.

The results presented in Table 2 demonstrate that FedGMKD consistently surpasses other federated learning techniques, especially under challenging heterogeneous data settings. Under the high heterogeneity scenario ($\beta = 0.2$), FedGMKD exhibits notable improvements. On CIFAR-10, it achieves the highest local accuracy of 74.34%, representing an increase of 3.4% over FedAvg, 1.93% over FedProto, and 2.26% over FPL. Moreover, FedGMKD attains the best global accuracy of 44.79%, outperforming FedProto and FPL by 7.43% and 2.72%, respectively. These significant improvements emphasize the ability of CKF and DAT to handle highly heterogeneous client data while capturing essential features. When the data is more homogeneous ($\beta = 5$), FedGMKD maintains its superior performance. For instance, it achieves the highest global accuracy of 53.84% on CIFAR-10, outperforming FedAvg by 1.12% and FPL by 0.19%, showcasing its effective model alignment even in near-homogeneous settings.

On the SVHN dataset, FedGMKD also demonstrates its robustness across varying heterogeneity levels. Under the high heterogeneity setting ($\beta = 0.2$), it achieves a local accuracy of 87.29%, outperforming FedAvg by 1.33% and FedProto by 0.71%. In terms of global accuracy, FedGMKD

Table 2: Performance of different schemes on CIFAR-10 and SVHN datasets under various data heterogeneity settings controlled by Dirichlet distribution parameter $\beta$.

| Scheme | CIFAR-10 | | | | SVHN | | | |
| | Local Acc | | Global Acc | | Local Acc | | Global Acc | |
| | $\beta = 0.2$ | $\beta = 5$ | $\beta = 0.2$ | $\beta = 5$ | $\beta = 0.2$ | $\beta = 5$ | $\beta = 0.2$ | $\beta = 5$ |
|---|---|---|---|---|---|---|---|---|
| FedAvg | 70.94% | 43.85% | 44.78% | 52.72% | 85.96% | 79.55% | 55.46% | 86.28% |
| FedProx | 71.15% | 43.72% | 44.59% | 52.56% | 86.08% | 80.01% | 56.36% | 86.41% |
| FedMD | 71.56% | 45.91% | 43.84% | 52.95% | 87.31% | 79.74% | **72.89%** | 86.29% |
| FedGen | 71.32% | 42.08% | 40.64% | 51.61% | 85.45% | 78.71% | 64.42% | 86.01% |
| Moon | 71.65% | 44.77% | 43.43% | 52.11% | 85.66% | 78.91% | 66.65% | 86.37% |
| FedProto | 72.41% | 46.12% | 37.36% | 52.81% | 86.58% | 79.84% | 69.31% | 86.81% |
| FPL | 72.08% | **47.11%** | 42.07% | 53.65% | 85.49% | 79.82% | 67.72% | 86.02% |
| **FedGMKD** | **74.34%** | 46.81% | **44.79%** | **53.84%** | **87.29%** | **80.41%** | 70.11% | **86.94%** |

shows a substantial improvement of 14.65% over FedAvg, with a global accuracy of 70.11%. Under the more homogeneous setting ($\beta = 5$), FedGMKD continues to deliver top performance, achieving a local accuracy of 80.41% and a global accuracy of 86.94%, outperforming FedProto by 0.57% in local accuracy and by 0.13% in global accuracy.

These results underscore the effectiveness of FedGMKD in addressing both extreme heterogeneity and more homogeneous environments. Its ability to consistently improve both local and global accuracy across different datasets highlights the robustness of the CKF and DAT techniques. By leveraging these mechanisms, FedGMKD captures high-quality local model representations while optimizing the aggregation process, ensuring better personalization and improved global performance in federated learning.

### A.4.2 Evaluation of Component Effectiveness in FedGMKD

In this section, we perform an ablation study to assess the contributions of the CKF and DAT within our FedGMKD framework. We compare the performance of FedGMKD with its variants—FedGMKD using only CKF and FedGMKD using only DAT—and a baseline method, FedAvg, on CIFAR-10 and SVHN datasets under different data heterogeneity settings, controlled by the Dirichlet distribution parameter $\beta$. The results are presented in Table 3.

Table 3: Performance of different schemes on CIFAR-10 and SVHN datasets under various data heterogeneity settings controlled by Dirichlet distribution parameter $\beta$.

| Scheme | CIFAR-10 | | | | SVHN | | | |
| | Local Acc | | Global Acc | | Local Acc | | Global Acc | |
| | $\beta = 0.2$ | $\beta = 5$ | $\beta = 0.2$ | $\beta = 5$ | $\beta = 0.2$ | $\beta = 5$ | $\beta = 0.2$ | $\beta = 5$ |
|---|---|---|---|---|---|---|---|---|
| FedAvg | 70.94 | 43.85 | 44.84 | 52.72 | 85.96 | 79.55 | 55.46 | 86.28 |
| FedGMKD (Only CKF) | 73.74 | 46.06 | 41.66 | 53.07 | 86.86 | 80.18 | 69.02 | 86.35 |
| FedGMKD (Only DAT) | 73.39 | 42.42 | 45.65 | 52.33 | 86.41 | 79.84 | 69.93 | 86.45 |
| FedGMKD | 74.34 | 46.81 | 44.79 | 53.84 | 87.29 | 80.41 | 70.11 | 86.84 |

Table 3 shows that FedGMKD, incorporating both CKF and DAT, consistently outperforms the baseline FedAvg method under various data heterogeneity conditions. On the CIFAR-10 dataset with high heterogeneity ($\beta = 0.2$), FedGMKD achieves an improvement of approximately 3.4% in local accuracy over FedAvg, while the variants utilizing only CKF and only DAT show improvements of 2.8% and 2.45%, respectively. Similarly, on the SVHN dataset under the same high heterogeneity, FedGMKD achieves a 1.33% increase in local accuracy and a substantial 14.65% increase in global accuracy over FedAvg. The CKF-only and DAT-only variants also show significant improvements over FedAvg.

For the CIFAR-10 dataset with high heterogeneity ($\beta = 0.2$), FedGMKD using only CKF achieves a local accuracy of 73.74%, while FedGMKD using only DAT reaches a global accuracy of 45.65%. These results indicate that CKF plays a crucial role in improving local model accuracy by effectively leveraging client-specific data characteristics, whereas DAT ensures that local improvements are

properly aggregated into the global model. On the SVHN dataset, CKF alone results in a local accuracy of 86.86% and a global accuracy of 69.02%, outperforming the DAT-only variant in local accuracy but not in global accuracy. This suggests that CKF is particularly beneficial in highly non-IID settings where local data characteristics are vital for model performance.

Nevertheless, the full integration of both CKF and DAT in FedGMKD consistently yields the highest performance across all metrics and settings. For example, on the SVHN dataset with $\beta = 0.2$, FedGMKD achieves a local accuracy of 87.29% and a global accuracy of 70.11%, outperforming both the CKF-only and DAT-only variants. This demonstrates that CKF and DAT work synergistically to enhance both local and global model performance, making FedGMKD robust and effective in diverse data environments.

### A.4.3   Impact of Model Complexity

In addition to our previous experiments, we conducted an ablation study to assess the impact of model complexity by using the ResNet-50 architecture in comparison with ResNet-18 on the CIFAR-10 dataset. The goal was to explore how a deeper network affects local and global model performance in the context of federated learning. Table 4 summarizes the performance of FedGMKD and other baseline methods using both ResNet-18 and ResNet-50.

Table 4: Comparison of performance for various schemes on CIFAR-10 using ResNet-18 and ResNet-50 architectures.

| Scheme | Acc (ResNet-18) | | Acc (ResNet-50) | |
|---|---|---|---|---|
| | Local Acc | Global Acc | Local Acc | Global Acc |
| FedAvg | 61.78 | 49.78 | 41.69 | 49.58 |
| FedProx | 64.04 | 55.16 | 43.25 | 49.67 |
| FedMD | 62.05 | 53.73 | 43.34 | 49.85 |
| FedGen | 60.17 | 51.55 | 42.81 | 48.99 |
| FedProto | 62.85 | 50.88 | 43.35 | 49.98 |
| Moon | 62.74 | 52.04 | 42.05 | 48.52 |
| FPL | 62.74 | 52.04 | 43.71 | 49.78 |
| **FedGMKD** | **65.69** | **60.31** | **46.27** | **50.48** |

As shown in Table 4, FedGMKD consistently outperforms other federated learning methods under both ResNet-18 and ResNet-50 configurations. However, there are noticeable differences in performance between the two architectures. For ResNet-50, while FedGMKD achieves the highest local and global accuracies among all methods, the performance does not exceed that of ResNet-18. Specifically, FedGMKD achieves a local accuracy of 46.27% and a global accuracy of 50.48% with ResNet-50, compared to 65.69% and 60.31%, respectively, with ResNet-18. The primary reason for the reduced performance of ResNet-50 in comparison to ResNet-18 lies in the increased model complexity and the corresponding challenges in federated learning environments. Deeper networks like ResNet-50 require more training epochs and greater communication bandwidth between the server and clients due to the larger number of parameters. The federated learning framework, particularly under constrained network conditions and limited communication rounds, may not be able to fully exploit the representational power of ResNet-50. Additionally, deeper models tend to require more iterations to converge, and given the fixed number of global communication rounds, ResNet-50 might not have had sufficient time to fully optimize in this federated setting.

While ResNet-50 offers higher theoretical representational power, practical considerations such as communication efficiency and the number of training rounds needed to achieve convergence limit its effectiveness in federated learning. FedGMKD with ResNet-50 still outperforms other schemes, demonstrating the robustness of our framework. However, the results suggest that balancing model complexity with communication and computational constraints is crucial in federated settings. Future work may explore strategies to mitigate these challenges, such as adaptive aggregation techniques and personalized model training strategies, to further improve the performance of deeper models like ResNet-50 in federated learning.

### A.4.4 Hyperparameter Exploration in FedGMKD

This section explores the effect of varying the regularization coefficients $\lambda$ and $\gamma$ within the FedGMKD framework on the CIFAR-10 dataset with 10 clients over 50 epochs. Table 5 summarizes the local and global accuracy for different combinations of $\lambda$ and $\gamma$, while also providing baseline comparisons with FedAvg, FedProto, and FPL.

Table 5: FedGMKD performance with varying $\lambda$ and $\gamma$ values on CIFAR-10 dataset (10 clients, 50 epochs).

| Scheme | $\lambda$ | $\gamma$ | Local Acc (%) | Global Acc (%) |
|---|---|---|---|---|
| FedGMKD | 0.06 | 0 | 60.14 | 48.17 |
| | 0.06 | 0.02 | 60.32 | 49.44 |
| | 0.06 | 0.04 | 60.99 | 49.48 |
| | 0.06 | 0.06 | **61.78** | **49.98** |
| | 0.06 | 0.08 | 60.27 | 48.64 |
| | 0.06 | 0.10 | 60.14 | 49.97 |
| | 0 | 0.06 | 60.73 | 48.72 |
| | 0.02 | 0.06 | 60.33 | 48.21 |
| | 0.04 | 0.06 | 61.01 | 49.47 |
| | 0.08 | 0.06 | 60.93 | 49.69 |
| | 0.10 | 0.06 | 59.86 | 47.52 |
| FedAvg | - | - | 55.75 | 46.62 |
| FedProto | - | - | 59.77 | 48.97 |
| FPL | - | - | 60.95 | 47.19 |

As shown in Table 5, FedGMKD demonstrates strong performance across a range of $\lambda$ and $\gamma$ values. The highest local and global accuracies are achieved with $\lambda = 0.06$ and $\gamma = 0.06$, resulting in a local accuracy of 61.78% and a global accuracy of 49.98%.

FedGMKD consistently outperforms the baseline methods FedAvg, FedProto, and FPL across all configurations, particularly at the optimal setting ($\lambda = 0.06$, $\gamma = 0.06$), where it achieves a local accuracy improvement of 6.03% and a global accuracy improvement of 3.36% over FedAvg. This indicates that FedGMKD's CKF and DAT mechanisms effectively handle Non-IID data and improve both local and global model performance.

The sensitivity analysis indicates that while FedGMKD performs well across different settings, fine-tuning $\lambda$ and $\gamma$ is crucial for maximizing performance. The robustness of FedGMKD is highlighted by its stable accuracy across various configurations, though the results suggest that balancing the feature alignment ($\lambda$) and prediction alignment ($\gamma$) terms is key to optimizing performance. This balance allows FedGMKD to maintain consistency between local and global models and handle diverse data distributions effectively.

### A.4.5 Comparison Between Hyper-Knowledge Averaging with DAT and FedGMKD

In this section, we present a comparison between an approach that utilizes the hyper-knowledge concept from FedHKD—specifically averaging features and soft predictions—and our proposed FedGMKD framework. Both methods incorporate the Discrepancy-Aware Aggregation Technique (DAT) across CIFAR-10, SVHN, and CIFAR-100 datasets with varying numbers of clients. Table 6 summarizes the local and global accuracy results for both methods, demonstrating the differences in their effectiveness when applied to non-IID federated learning scenarios.

As shown in Table 6, FedGMKD consistently outperforms the method based on hyper-knowledge averaging with DAT across all datasets and client configurations. The GMM-based approach in FedGMKD allows for the extraction of more precise prototype features and soft predictions by clustering data points according to their similarity. This clustering technique captures the inherent structure and diversity within the data better than the simple averaging of features and soft predictions derived from hyper-knowledge. By leveraging these GMM-derived prototypes, FedGMKD provides a richer and more representative set of features for aggregation, leading to superior performance in both local and global metrics.

Table 6: Comparison of Hyper-Knowledge Averaging with DAT and FedGMKD on CIFAR-10, SVHN, and CIFAR-100 datasets with $\beta = 0.5$.

| Dataset | Clients | FedGMKD Acc (%) | | HK with DAT Acc (%) | |
|---------|---------|-------|--------|-------|--------|
| | | Local | Global | Local | Global |
| CIFAR-10 | 10 | 61.78 | 49.78 | 60.33 | 48.96 |
| | 20 | 64.04 | 55.16 | 62.31 | 52.67 |
| | 50 | 65.69 | 60.31 | 64.23 | 58.64 |
| SVHN | 10 | 86.26 | 82.64 | 85.91 | 81.45 |
| | 20 | 87.43 | 87.78 | 86.75 | 87.69 |
| | 50 | 87.16 | 90.17 | 86.98 | 89.78 |
| CIFAR-100 | 10 | 17.16 | 16.97 | 16.12 | 16.09 |
| | 20 | 20.96 | 21.56 | 19.71 | 20.69 |
| | 50 | 23.57 | 24.63 | 22.44 | 23.82 |

Furthermore, while both methods use the discrepancy-aware aggregation technique (DAT) to weight client contributions, FedGMKD's clustering-based feature extraction aligns more closely with client data distributions. This alignment ensures that the global model benefits from a more accurate and relevant integration of client data, particularly in highly heterogeneous settings. The close performance in global accuracy between the two methods in some configurations shows that DAT effectively mitigates client discrepancies; however, the advantage of using GMMs for prototype generation becomes evident as FedGMKD achieves consistently higher global accuracy, demonstrating the effectiveness of its integrated CKF and DAT mechanisms.

### A.4.6 Evaluation on IMDB Dataset

To further assess the adaptability of FedGMKD across different data modalities, we conducted experiments on the IMDB (Internet Movie Database) dataset, a prominent benchmark in natural language processing (NLP) used for sentiment analysis tasks. The IMDB dataset comprises a large collection of movie reviews, presenting a significant challenge for federated learning due to the inherent diversity of client data distributions. Sentiment analysis in particular requires models to generalize effectively across these distributed datasets.

The experimental setup mirrored the structure of previous evaluations, utilizing 10 clients over the course of 50 training epochs. As the task shifted from computer vision to NLP, the model architecture was updated from ResNet-18 to BERT (Bidirectional Encoder Representations from Transformers), a model widely recognized for its effectiveness in text-based tasks like sentiment analysis. BERT's ability to capture contextual embeddings makes it particularly well-suited for handling the complexity of the IMDB dataset. Evaluation metrics included local accuracy, global accuracy, and the average computation time per client.

Table 7: Performance of different schemes on IMDB dataset using BERT model (10 clients, 50 epochs).

| Scheme | Local Acc (%) | Global Acc (%) | Avg Time (s) |
|--------|---------------|----------------|--------------|
| FedAvg | 83.71 | 50.52 | 411.95 |
| FedProx | 83.75 | 48.50 | 438.52 |
| FedMD | 83.87 | 48.29 | 700.73 |
| FedGen | 83.54 | 49.16 | 471.35 |
| FedProto | 84.13 | 49.72 | 586.77 |
| FPL | 83.96 | 50.12 | 665.29 |
| **FedGMKD** | **85.11** | **51.58** | 677.79 |

Table 7 shows that FedGMKD consistently outperforms other federated learning algorithms in terms of both local and global accuracy on the IMDB dataset. FedGMKD achieves the highest local accuracy of 85.11% and the highest global accuracy of 51.58%, indicating its robustness in handling

non-vision datasets. In particular, the results highlight the effectiveness of CKF and DAT in adapting to text-based tasks. While the average computation time for FedGMKD is slightly higher due to the complexity of BERT, the gains in both local and global accuracy justify the computational overhead. These findings underscore the versatility of FedGMKD, demonstrating that it can generalize well to NLP tasks and effectively handle different data modalities, further validating its broader applicability across federated learning scenarios.

### A.5 Feature Representations

In this section, we present an ablation study comparing the t-SNE visualizations of feature representations obtained using different federated learning methods: FedAvg, FedProto, FPL, and FedGMKD. These experiments were conducted with 10 clients and a data heterogeneity parameter ($\beta$) of 0.5 on the CIFAR-10 dataset. The primary aim is to evaluate how effectively each method learns to separate feature representations by class, as visualized using t-SNE.

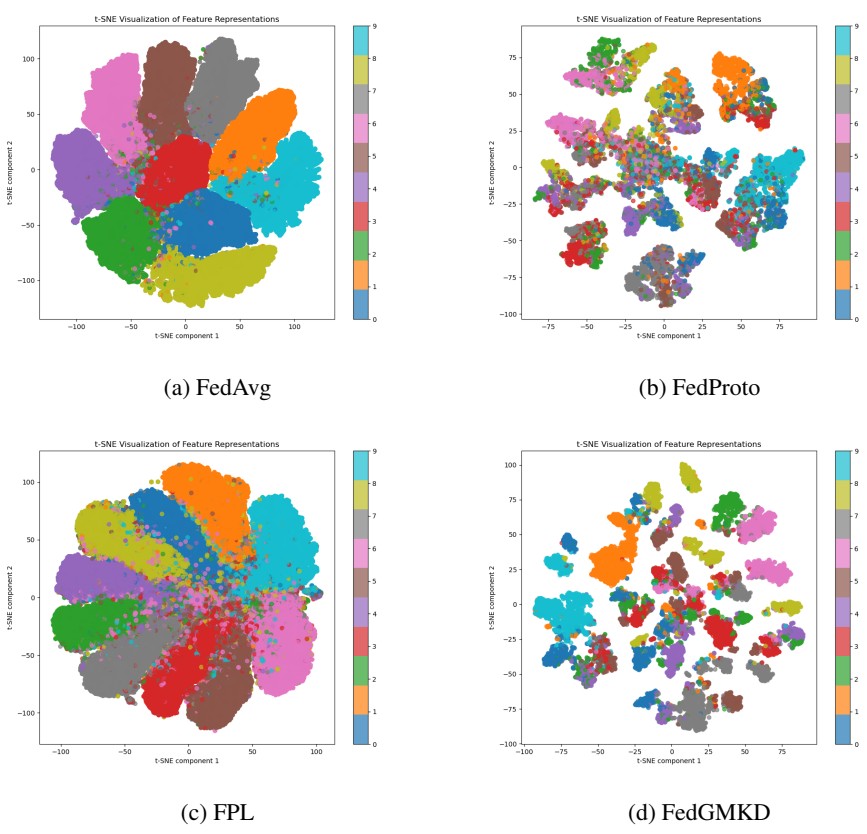

Figure 4: Qualitative comparison of t-SNE visualization among FedAvg, FedProto, FPL and FedGMKD. Compared with other methods, the feature distribution of the FedGMKD is more compact within each category, and more discriminative across classes.

In the FedAvg method (Figure 4a), feature representations are widely dispersed with significant overlap between classes. Although FedAvg captures some class-specific features, the resulting feature clusters are neither distinct nor compact, indicating weak separation between the classes. This is expected, as FedAvg primarily averages model parameters without specific focus on class differentiation or feature compactness.

The FedProto method (Figure 4b), while not explicitly clustering features, introduces prototype-based learning where class prototypes guide the learning process. The t-SNE visualization shows slightly more distinct feature distributions compared to FedAvg, reflecting an improvement in feature separation. However, since FedProto emphasizes prototyping over clustering, some overlap between class features still persists, suggesting room for more effective inter-class separation.

FPL (Figure 4c) shows clustering behavior very similar to FedAvg, with large overlaps between clusters and less distinct boundaries. The feature separation is minimal, and the clusters are not as compact as those in FedProto. This indicates that FPL struggles to produce clear, discriminative class-specific features, making its clustering quality less ideal compared to FedProto.

FedGMKD (Figure 4d) demonstrates the most compact and well-separated feature representations. The t-SNE visualization shows that FedGMKD's CKF successfully generates class-specific prototypes that contribute to tight and distinct feature clusters. This clear separation highlights how CKF, along with the DAT, enhances feature learning across clients, leading to superior class separation and making FedGMKD the most effective method in terms of feature representation quality.

### A.6 Convergence Analysis of FedGMKD

#### A.6.1 Assumptions

To prove the convergence of FedGMKD, we introduce the following assumptions:

**Assumption 1 (Lipschitz Continuity of the Local Loss Function):** The local loss function $L(\mathcal{D}_i, \mathbf{w}_i^r)$ at each client $i$ is $L$-smooth, meaning that its gradient is Lipschitz continuous with a constant $L > 0$. Formally, for any $\mathbf{w}_i^a, \mathbf{w}_i^b$, we have:

$$\|\nabla L(\mathcal{D}_i, \mathbf{w}_i^a) - \nabla L(\mathcal{D}_i, \mathbf{w}_i^b)\| \leq L\|\mathbf{w}_i^a - \mathbf{w}_i^b\|. \tag{16}$$

This ensures that the local loss function is smooth, and the gradient changes gradually for any two parameter vectors.

**Assumption 2 (Unbiased Gradient and Bounded Variance):** The gradient estimation at each client $i$ is unbiased with bounded variance. Specifically, for any $\mathbf{w}_i^r$, the expectation of the local gradient equals the true global gradient at the global parameter $\mathbf{w}^R$, and the variance of the gradient is bounded by a constant $\sigma^2$:

$$\mathbb{E}[\nabla L(\mathcal{D}_i, \mathbf{w}_i^r)] = \nabla L(\mathbf{w}^R), \tag{17}$$

and

$$\mathbb{E}[\|\nabla L(\mathcal{D}_i, \mathbf{w}_i^r) - \nabla L(\mathbf{w}^R)\|^2] \leq \sigma^2. \tag{18}$$

**Assumption 3 (Bounded Gradient Norm):** The gradient of the local loss function $L(\mathcal{D}_i, \mathbf{w}_i^r)$ is bounded. There exists a constant $G > 0$ such that:

$$\|\nabla L(\mathcal{D}_i, \mathbf{w}_i^r)\| \leq G, \quad \forall i, r. \tag{19}$$

**Assumption 4 (Bounded Discrepancy in KL Divergence):** The Kullback-Leibler (KL) divergence between the local model's predictions and the global model's predictions for each class is bounded. For any client $i$ and class $j$, the KL divergence $d_i^j$ between the local soft prediction $\hat{q}_i^j$ and the global soft prediction $\hat{Q}_j$ satisfies:

$$d_i^j = D_{\text{KL}}(\hat{q}_i^j \parallel \hat{Q}_j) \leq D_{\text{KL}}^{\max}. \tag{20}$$

This ensures that the discrepancy between local and global models is controlled.

**Assumption 5 (Cluster Knowledge Fusion Convergence):** The Gaussian Mixture Model (GMM)-based clustering used in Cluster Knowledge Fusion (CKF) converges to a stable clustering of client updates. Let $M$ be the number of Gaussian components (to avoid conflict with global round notation), and let $\gamma_m(\mathbf{x}_i^j)$ represent the responsibility of the $m$-th Gaussian component for a data point $\mathbf{x}_i^j$. After a sufficient number of iterations, we assume:

$$\sum_{m=1}^{M} \gamma_m(\mathbf{x}_i^j) = 1 \quad \text{and} \quad \hat{h}_i^j = \sum_{m=1}^{M} \gamma_m(\mathbf{h}_i^j)^-{}_{m_j}. \tag{21}$$

#### A.6.2 Lemma

## Lemma 1

**Lemma 1:** Under Assumptions 1-5 A.6.1, after $R$ local training epochs at global round $r + 1$, the local loss function can be bounded as follows:

$$\mathbb{E}[L(\mathcal{D}_i, \mathbf{w}_i^{r+R})] \leq L(\mathcal{D}_i, \mathbf{w}_i^r) - \eta R \mathbb{E}[\|\nabla L(\mathcal{D}_i, \mathbf{w}_i^r)\|^2] + \frac{L\eta^2 RG^2}{2} + \frac{\eta^2\sigma^2}{2}, \tag{22}$$

where $L$ is the Lipschitz constant from Assumption 1, $\sigma^2$ is the variance bound from Assumption 2, $G$ is the bound on the gradient norm from Assumption 3, and $\eta$ is the learning rate.

**Proof:**

In each communication round $r$, client $i$ performs $R$ local updates based on stochastic gradient descent (SGD) on its local dataset $\mathcal{D}_i$. The local training objective for client $i$ is defined as:

$$
L(\mathcal{D}_i, \mathbf{w}_i) = \frac{1}{|\mathcal{D}_i|} \sum_{(x_k, y_k) \in \mathcal{D}_i} \ell\left(C_{\psi_i}\left(F_{\theta_i}(x_k)\right), y_k\right)
$$
$$
+ \lambda \frac{1}{|\mathcal{D}_i|} \sum_{(x_k, y_k) \in \mathcal{D}_i} \left\| F_{\theta_i}(x_k) - \mathbf{H}_{y_k}^{r+1} \right\|_2^2 + \frac{\gamma}{n} \sum_{j=1}^{n} \left\| \frac{C_{\psi_i}\left(\mathbf{H}_j^{r+1}\right)}{T} - \frac{\mathbf{Q}_j^{r+1}}{T} \right\|_2^2. \tag{23}
$$

First, we use the Lipschitz continuity of the gradient (Assumption 1). This assumption implies that the loss function $L(\mathcal{D}_i, \mathbf{w})$ is $L$-smooth. Formally:

$$
L(\mathcal{D}_i, \mathbf{w}_1) \leq L(\mathcal{D}_i, \mathbf{w}_2) + \nabla L(\mathcal{D}_i, \mathbf{w}_2)^T(\mathbf{w}_1 - \mathbf{w}_2) + \frac{L}{2}\|\mathbf{w}_1 - \mathbf{w}_2\|^2. \tag{24}
$$

We set $\mathbf{w}_1 = \mathbf{w}_i^{r+j+1}$ and $\mathbf{w}_2 = \mathbf{w}_i^{r+j}$. Substituting these into the Lipschitz continuity condition:

$$
L(\mathcal{D}_i, \mathbf{w}_i^{r+j+1}) \leq L(\mathcal{D}_i, \mathbf{w}_i^{r+j}) + \nabla L(\mathcal{D}_i, \mathbf{w}_i^{r+j})^T(\mathbf{w}_i^{r+j+1} - \mathbf{w}_i^{r+j}) + \frac{L}{2}\|\mathbf{w}_i^{r+j+1} - \mathbf{w}_i^{r+j}\|^2. \tag{25}
$$

The SGD update rule is given by:

$$
\mathbf{w}_i^{r+j+1} = \mathbf{w}_i^{r+j} - \eta \nabla L(\mathcal{D}_i, \mathbf{w}_i^{r+j}), \tag{26}
$$

where $\eta$ is the learning rate. Substituting this into the previous equation:

$$
L(\mathcal{D}_i, \mathbf{w}_i^{r+j+1}) \leq L(\mathcal{D}_i, \mathbf{w}_i^{r+j}) - \eta\|\nabla L(\mathcal{D}_i, \mathbf{w}_i^{r+j})\|^2 + \frac{L\eta^2}{2}\|\nabla L(\mathcal{D}_i, \mathbf{w}_i^{r+j})\|^2. \tag{27}
$$

To analyze the expected change in the loss, we take the expectation on both sides with respect to the randomness of the gradient:

$$
\mathbb{E}\left[L(\mathcal{D}_i, \mathbf{w}_i^{r+j+1})\right] \leq \mathbb{E}\left[L(\mathcal{D}_i, \mathbf{w}_i^{r+j})\right] - \eta \mathbb{E}\left[\|\nabla L(\mathcal{D}_i, \mathbf{w}_i^{r+j})\|^2\right] + \frac{L\eta^2}{2}\mathbb{E}\left[\|\nabla L(\mathcal{D}_i, \mathbf{w}_i^{r+j})\|^2\right]. \tag{28}
$$

According to Assumption 2 (Unbiased Gradient and Bounded Variance), the gradient is an unbiased estimator of the true gradient, and its variance is bounded:

$$
\mathbb{E}\left[\|\nabla L(\mathcal{D}_i, \mathbf{w}_i^{r+j}) - \nabla L(\mathbf{w})\|^2\right] \leq \sigma^2. \tag{29}
$$

We decompose the expected squared gradient norm as follows:

$$
\mathbb{E}\left[\|\nabla L(\mathcal{D}_i, \mathbf{w}_i^{r+j})\|^2\right] = \|\nabla L(\mathcal{D}_i, \mathbf{w}_i^{r+j})\|^2 + \mathrm{Var}\left(\nabla L(\mathcal{D}_i, \mathbf{w}_i^{r+j})\right). \tag{30}
$$

Thus:

$$
\mathbb{E}\left[\|\nabla L(\mathcal{D}_i, \mathbf{w}_i^{r+j})\|^2\right] \leq \|\nabla L(\mathcal{D}_i, \mathbf{w}_i^{r+j})\|^2 + \sigma^2. \tag{31}
$$

To understand the cumulative effect of $R$ local updates, we sum the inequalities over $j = 0$ to $j = R - 1$:

$$
\mathbb{E}[L(\mathcal{D}_i, \mathbf{w}_i^{r+R})] \leq L(\mathcal{D}_i, \mathbf{w}_i^{r}) - \eta \sum_{j=0}^{R-1} \mathbb{E}[\|\nabla L(\mathcal{D}_i, \mathbf{w}_i^{r+j})\|^2] + \frac{L\eta^2}{2} \sum_{j=0}^{R-1} \mathbb{E}[\|\nabla L(\mathcal{D}_i, \mathbf{w}_i^{r+j})\|^2]. \tag{32}
$$

From Assumption 3 (Bounded Gradient Norm), $\|\nabla L(\mathcal{D}_i, \mathbf{w}_i^{r+j})\| \leq G$ for all $j$. Substituting this bound:

$$\sum_{j=0}^{R-1} \mathbb{E}[\|\nabla L(\mathcal{D}_i, \mathbf{w}_i^{r+j})\|^2] \leq R(G^2 + \sigma^2). \tag{33}$$

Substituting this result into the inequality:

$$\mathbb{E}[L(\mathcal{D}_i, \mathbf{w}_i^{r+R})] \leq L(\mathcal{D}_i, \mathbf{w}_i^r) - \eta R \mathbb{E}[\|\nabla L(\mathcal{D}_i, \mathbf{w}_i^r)\|^2] + \frac{L\eta^2 R(G^2 + \sigma^2)}{2}. \tag{34}$$

## Lemma 2

**Lemma 2:** Under Assumptions 1-5 A.6.1, the global loss function after one global aggregation step at round $r + 1$ can be bounded as:

$$\mathbb{E}[F(\mathbf{W}^{r+1})] \leq F(\mathbf{W}^r) - \eta R \left(1 - \frac{L\eta}{2}\right) \sum_{i=1}^n w_i' \mathbb{E}\left[\|\nabla F_i(\mathbf{w}_i^r)\|^2\right] + \frac{L\eta^2 \sigma^2}{2}, \tag{35}$$

where $F(\mathbf{W})$ is the global loss function, $F_i(\mathbf{w}_i)$ is the local loss function for client $i$, $w_i'$ are the discrepancy-aware weights, and $\sigma^2$ is the variance bound from Assumption 2.

**Proof:**

The global model after aggregation at round $r + 1$ is computed as:

$$\mathbf{W}^{r+1} = \sum_{i=1}^n w_i' \mathbf{w}_i^{r+R}, \tag{36}$$

where $w_i'$ are the discrepancy-aware weights based on each client's data quality and quantity, and $\mathbf{w}_i^{r+R}$ represents the model parameters after $R$ local updates for client $i$.

We express the global loss function at time $r + 1$ in terms of the aggregated client models:

$$F(\mathbf{W}^{r+1}) = \sum_{i=1}^n w_i' F_i(\mathbf{w}_i^{r+R}), \tag{37}$$

where $F_i(\mathbf{w}_i^{r+R})$ is the local loss function value for client $i$ after $R$ local updates.

From Lemma 1, we have:

$$\mathbb{E}\left[F_i(\mathbf{w}_i^{r+R})\right] \leq F_i(\mathbf{w}_i^r) - \eta R \rho \mathbb{E}\left[\|\nabla F_i(\mathbf{w}_i^r)\|^2\right] + \frac{L\eta^2 R(G^2 + \sigma^2)}{2}, \tag{38}$$

where $\rho$ represents the cosine similarity between the gradients and $L$, $\sigma^2$, and $G$ are constants derived from the assumptions.

Taking the expectation over all clients and substituting the bound from Lemma 1:

$$\mathbb{E}[F(\mathbf{W}^{r+1})] = \sum_{i=1}^n w_i' \mathbb{E}\left[F_i(\mathbf{w}_i^{r+R})\right]. \tag{39}$$

Substituting the inequality from Lemma 1 into this expression:

$$\mathbb{E}[F(\mathbf{W}^{r+1})] \leq \sum_{i=1}^n w_i' \left[F_i(\mathbf{w}_i^r) - \eta R \rho \mathbb{E}\left[\|\nabla F_i(\mathbf{w}_i^r)\|^2\right] + \frac{L\eta^2 R(G^2 + \sigma^2)}{2}\right]. \tag{40}$$

Recall that the global loss function at time $r$ is:

$$F(\mathbf{W}^r) = \sum_{i=1}^n w_i' F_i(\mathbf{w}_i^r). \tag{41}$$

Substituting this into the inequality:

$$\mathbb{E}[F(\mathbf{W}^{r+1})] \le F(\mathbf{W}^r) - \eta R \rho \sum_{i=1}^{n} w_i' \mathbb{E}\left[\|\nabla F_i(\mathbf{w}_i^r)\|^2\right] + \frac{L\eta^2 R \sum_{i=1}^{n} w_i'(G^2 + \sigma^2)}{2}. \quad (42)$$

According to Assumption 2 (Unbiased Gradient and Bounded Variance), we can further refine the variance term:

$$\sum_{i=1}^{n} w_i' \sigma^2 = \sigma^2, \quad (43)$$

since the weights $w_i'$ sum to 1. Therefore:

$$\mathbb{E}[F(\mathbf{W}^{r+1})] \le F(\mathbf{W}^r) - \eta R \rho \sum_{i=1}^{n} w_i' \mathbb{E}\left[\|\nabla F_i(\mathbf{w}_i^r)\|^2\right] + \frac{L\eta^2 \sigma^2}{2}. \quad (44)$$

The factor $\rho$ represents the alignment of the gradient directions across clients. In practice, $\rho$ can vary, but for worst-case analysis, we set $\rho = 1$ to provide a conservative bound. Substituting $\rho = 1$:

$$\mathbb{E}[F(\mathbf{W}^{r+1})] \le F(\mathbf{W}^r) - \eta R \sum_{i=1}^{n} w_i' \mathbb{E}\left[\|\nabla F_i(\mathbf{w}_i^r)\|^2\right] + \frac{L\eta^2 \sigma^2}{2}. \quad (45)$$

We rearrange the expression to factor out the learning rate $\eta$ and the Lipschitz constant $L$:

$$\mathbb{E}[F(\mathbf{W}^{r+1})] \le F(\mathbf{W}^r) - \eta R \left(1 - \frac{L\eta}{2}\right) \sum_{i=1}^{n} w_i' \mathbb{E}\left[\|\nabla F_i(\mathbf{w}_i^r)\|^2\right] + \frac{L\eta^2 \sigma^2}{2}. \quad (46)$$

### A.6.3 Theorem

## Theorem 1: FedGMKD Convergence

Under Assumptions 1-5 A.6.1, for any client $i$, after $R$ global communication rounds, the expected global loss function is bounded as:

$$\frac{1}{R} \sum_{r=1}^{R} \sum_{i=1}^{n} w_i' \mathbb{E}\left[\|\nabla F_i(\mathbf{w}_i^r)\|^2\right] \le \frac{F(\mathbf{W}^1) - F^*}{\eta R^2} + \sigma^2 + \frac{L\eta R G^2}{2}, \quad (47)$$

where $F(\mathbf{W})$ is the global loss function, $F^*$ is the lower bound of the global objective function, $\eta$ is the learning rate, $R$ is the number of global updates, $L$ is the Lipschitz constant from Assumption 1, $\sigma^2$ is the variance bound on the gradient from Assumption 2, and $G$ is the upper bound on the gradient norm from Assumption 3.

As $R \to \infty$, the global loss function converges to a neighborhood around a stationary point, with the size of the neighborhood bounded by:

$$\sigma^2 + \frac{L\eta R G^2}{2}. \quad (48)$$

**Proof:**

The global model update after aggregation at round $r + 1$ is defined as:

$$\mathbf{W}^{r+1} = \sum_{i=1}^{n} w_i' \mathbf{w}_i^{r+R}, \quad (49)$$

where $w_i'$ are the discrepancy-aware weights, and $\mathbf{w}_i^{r+R}$ are the parameters of client $i$ after $R$ local updates.

The global loss function after aggregation is:

$$F(\mathbf{W}^{r+1}) = \sum_{i=1}^{n} w_i' F_i(\mathbf{w}_i^{r+R}), \quad (50)$$

where $F_i(\mathbf{w}_i^{r+R})$ denotes the local loss function of client $i$ after $R$ local updates.

From Lemma 2, we know:

$$\mathbb{E}[F(\mathbf{W}^{r+1})] \leq F(\mathbf{W}^r) - \eta R \left(1 - \frac{L\eta}{2}\right) \sum_{i=1}^{n} w_i' \mathbb{E}\left[\|\nabla F_i(\mathbf{w}_i^r)\|^2\right] + \frac{L\eta^2 \sigma^2}{2}. \tag{51}$$

Let $\Delta_r$ denote the change in the global loss function:

$$\Delta_r = F(\mathbf{W}^r) - \mathbb{E}[F(\mathbf{W}^{r+1})]. \tag{52}$$

Substituting the result from Lemma 2:

$$\Delta_r \geq \eta R \left(1 - \frac{L\eta}{2}\right) \sum_{i=1}^{n} w_i' \mathbb{E}\left[\|\nabla F_i(\mathbf{w}_i^r)\|^2\right] - \frac{L\eta^2 \sigma^2}{2}. \tag{53}$$

This inequality indicates the amount by which the global loss function decreases in expectation after one round of global aggregation.

We sum the inequality $\Delta_r \geq \ldots$ over $r = 1$ to $R$:

$$\sum_{r=1}^{R} \Delta_r = F(\mathbf{W}^1) - \mathbb{E}[F(\mathbf{W}^{R+1})]. \tag{54}$$

Substituting the inequality for $\Delta_r$:

$$F(\mathbf{W}^1) - \mathbb{E}[F(\mathbf{W}^{R+1})] \geq \eta R \left(1 - \frac{L\eta}{2}\right) \sum_{r=1}^{R} \sum_{i=1}^{n} w_i' \mathbb{E}\left[\|\nabla F_i(\mathbf{w}_i^r)\|^2\right] - \frac{L\eta^2 \sigma^2 R}{2}. \tag{55}$$

Rearrange the inequality to isolate the sum of the expected gradient norms:

$$\eta R \left(1 - \frac{L\eta}{2}\right) \sum_{r=1}^{R} \sum_{i=1}^{n} w_i' \mathbb{E}\left[\|\nabla F_i(\mathbf{w}_i^r)\|^2\right] \leq F(\mathbf{W}^1) - \mathbb{E}[F(\mathbf{W}^{R+1})] + \frac{L\eta^2 \sigma^2 R}{2}. \tag{56}$$

To find the average over $R$ rounds, divide both sides by $\eta R \left(1 - \frac{L\eta}{2}\right) R$:

$$\frac{1}{R} \sum_{r=1}^{R} \sum_{i=1}^{n} w_i' \mathbb{E}\left[\|\nabla F_i(\mathbf{w}_i^r)\|^2\right] \leq \frac{F(\mathbf{W}^1) - \mathbb{E}[F(\mathbf{W}^{R+1})]}{\eta R \left(1 - \frac{L\eta}{2}\right) R} + \frac{L\eta \sigma^2}{2 \left(1 - \frac{L\eta}{2}\right)}. \tag{57}$$

Assuming that $F(\mathbf{W})$ is bounded below by $F^*$, such that:

$$F(\mathbf{W}^{R+1}) \geq F^*, \tag{58}$$

we substitute this bound into the inequality:

$$\frac{1}{R} \sum_{r=1}^{R} \sum_{i=1}^{n} w_i' \mathbb{E}\left[\|\nabla F_i(\mathbf{w}_i^r)\|^2\right] \leq \frac{F(\mathbf{W}^1) - F^*}{\eta R \left(1 - \frac{L\eta}{2}\right) R} + \frac{L\eta \sigma^2}{2 \left(1 - \frac{L\eta}{2}\right)}. \tag{59}$$

As $R \to \infty$, the term $\frac{F(\mathbf{W}^1) - F^*}{\eta R \left(1 - \frac{L\eta}{2}\right) R}$ tends to zero. Therefore:

$$\lim_{R \to \infty} \frac{1}{R} \sum_{r=1}^{R} \sum_{i=1}^{n} w_i' \mathbb{E}\left[\|\nabla F_i(\mathbf{w}_i^r)\|^2\right] \leq \frac{L\eta \sigma^2}{2 \left(1 - \frac{L\eta}{2}\right)}. \tag{60}$$

Since $\eta$ is chosen such that $1 - \frac{L\eta}{2} > 0$, this shows that the global loss function converges to a neighborhood around a stationary point with the size of the neighborhood given by:

$$\sigma^2 + \frac{L\eta R G^2}{2}. \tag{61}$$

## Theorem 2: FedGMKD Convergence Rate

Under Assumptions 1-5 A.6.1, for any client $i$, after $R$ global communication rounds, the convergence rate of the global loss function $F(\mathbf{W})$ is bounded as follows:

$$F(\mathbf{W}^R) - F^* \leq \frac{C_1}{R} + C_2, \tag{62}$$

where $F(\mathbf{W})$ is the global loss function, $F^*$ represents the lower bound of $F(\mathbf{W})$, and $C_1$ and $C_2$ are constants that depend on variance $\sigma^2$, Lipschitz constant $L$, learning rate $\eta$, and the number of local steps.

As $R \to \infty$, the global loss function converges to a neighborhood around a stationary point.

**Proof:**

To establish the convergence rate of $F(\mathbf{W})$, we begin with Lemma 2:

$$\mathbb{E}[F(\mathbf{W}^{r+1})] \leq F(\mathbf{W}^r) - \eta \left(1 - \frac{L\eta}{2}\right) \sum_{i=1}^{n} w_i' \mathbb{E}[\|\nabla F_i(\mathbf{w}_i^r)\|^2] + \frac{L\eta^2\sigma^2}{2}. \tag{63}$$

Define the decrease in the global loss function after one global round as:

$$\Delta_r = F(\mathbf{W}^r) - \mathbb{E}[F(\mathbf{W}^{r+1})]. \tag{64}$$

Substituting the result from Lemma 2:

$$\Delta_r \geq \eta \left(1 - \frac{L\eta}{2}\right) \sum_{i=1}^{n} w_i' \mathbb{E}[\|\nabla F_i(\mathbf{w}_i^r)\|^2] - \frac{L\eta^2\sigma^2}{2}. \tag{65}$$

This equation indicates that the decrease in the global loss function $F(\mathbf{W})$ after each round is proportional to the squared gradient norm and controlled by the variance and Lipschitz constant.

We sum $\Delta_r$ over $r = 1$ to $R$:

$$\sum_{r=1}^{R} \Delta_r = F(\mathbf{W}^1) - \mathbb{E}[F(\mathbf{W}^{R+1})]. \tag{66}$$

Substituting the inequality for $\Delta_r$:

$$F(\mathbf{W}^1) - \mathbb{E}[F(\mathbf{W}^{R+1})] \geq \eta \left(1 - \frac{L\eta}{2}\right) \sum_{r=1}^{R} \sum_{i=1}^{n} w_i' \mathbb{E}[\|\nabla F_i(\mathbf{w}_i^r)\|^2] - \frac{L\eta^2\sigma^2 R}{2}. \tag{67}$$

This inequality now relates the total decrease in the global loss function to the sum of the squared gradients over all rounds.

We rearrange the equation to isolate the sum of the expected gradient norms:

$$\eta \left(1 - \frac{L\eta}{2}\right) \sum_{r=1}^{R} \sum_{i=1}^{n} w_i' \mathbb{E}[\|\nabla F_i(\mathbf{w}_i^r)\|^2] \leq F(\mathbf{W}^1) - F^* + \frac{L\eta^2\sigma^2 R}{2}. \tag{68}$$

Dividing by $\eta \left(1 - \frac{L\eta}{2}\right) R$ We divide both sides of the inequality by $\eta \left(1 - \frac{L\eta}{2}\right) R$ to find the average squared gradient norm over $R$ rounds:

$$\frac{1}{R} \sum_{r=1}^{R} \sum_{i=1}^{n} w_i' \mathbb{E}[\|\nabla F_i(\mathbf{w}_i^r)\|^2] \leq \frac{F(\mathbf{W}^1) - F^*}{\eta \left(1 - \frac{L\eta}{2}\right) R} + \frac{L\eta\sigma^2}{2\left(1 - \frac{L\eta}{2}\right)}. \tag{69}$$

Define $C_1 = \frac{F(\mathbf{W}^1) - F^*}{\eta\left(1 - \frac{L\eta}{2}\right)}$ and $C_2 = \frac{L\eta\sigma^2}{2\left(1 - \frac{L\eta}{2}\right)}$. Substituting these constants:

$$\frac{1}{R} \sum_{r=1}^{R} \sum_{i=1}^{n} w_i' \mathbb{E}[\|\nabla F_i(\mathbf{w}_i^r)\|^2] \leq \frac{C_1}{R} + C_2. \tag{70}$$

As $R \to \infty$, the term $\frac{C_1}{R} \to 0$. This indicates that the average squared gradient norm converges to $C_2$.

Given that $C_2$ is bounded, the behavior of $F(\mathbf{W})$ over $R$ rounds implies:

$$F(\mathbf{W}^R) - F^* \leq \frac{C_1}{R} + C_2. \tag{71}$$

This shows that the global loss function $F(\mathbf{W})$ converges at a rate proportional to $\frac{1}{R}$, plus an asymptotic constant $C_2$ that defines the neighborhood around the stationary point.

As $R \to \infty$, the term $\frac{C_1}{R}$ diminishes, and the global model converges to a neighborhood around the stationary point defined by $C_2$.

