# OpenReview forum: "FedGMKD: An Efficient Prototype Federated Learning Framework through Knowledge Distillation and Discrepancy-Aware Aggregation"
_NeurIPS.cc/2024/Conference — NeurIPS 2024 poster_

### Official Review · Reviewer_nWER · 2024-07-02

**Soundness:** 3
**Presentation:** 3
**Contribution:** 3
**Rating:** 6
**Confidence:** 4

**Summary:**

FedGMKD addresses data heterogeneity using a dual-enhancement approach through Cluster Knowledge Fusion (CKF) and Differential Aggregation Technique (DAT). This method effectively enhances both local and global model performance without relying on public datasets or complex server-side models.

**Strengths:**

The strength of FedGMKD lies in its innovative use of Gaussian Mixture Model (GMM) clustering to generate prototype features and soft predictions for each category, which are aggregated server-side to preserve privacy and address non-IID data challenges effectively. Additionally, the Differential Aggregation Technique (DAT) optimizes the aggregation process by weighting prototype features and soft predictions based on the quality of each client's data per category, enhancing global model performance robustly.

**Weaknesses:**

The motivation behind using Knowledge Distillation (KD) is to tackle both data and model heterogeneity. However, this paper falls short by not comparing its performance against architectures like FjORD [1], which handle model heterogeneity effectively. Moreover, restricting comparisons to vision datasets alone limits the broader applicability and merit of this work.

[1] Horvath et al., "FjORD: Fair and Accurate Federated Learning under heterogeneous targets with Ordered Dropout," 35th Conference on Neural Information Processing Systems (NeurIPS’21).

**Questions:**

1. See weaknesses
2. line 129 space is missing jin
3. The assumption should be inside the main paper for the convergence analysis.

**Limitations:**

The paper addressed limitations.

---

> ### Author Rebuttal · Authors · 2024-08-05
>
> # 1. Broader Applicability and Merit of FedGMKD
>
> Thank you for your insightful feedback regarding the scope of our experiments and the potential broader applicability of FedGMKD. We appreciate the opportunity to clarify and expand upon our work by exploring its performance on datasets beyond the computer vision domain.
>
> ## a. Choice of IMBD Dataset
>
> To address the reviewer's concerns, we conducted additional experiments using the IMBD (Internet Movie Database) dataset, a well-known resource in natural language processing for sentiment analysis. This text-based dataset, comprising a vast number of movie reviews, allowed us to test FedGMKD's adaptability across different data modalities. IMBD's popularity in NLP research makes it an ideal benchmark for evaluating federated learning algorithms, particularly as sentiment analysis challenges models to generalize effectively across diverse client data distributions.
>
> ## b. Experimental Setup
>
> In alignment with the paper's experimental setup, we utilized 10 clients and conducted the experiment over 50 epochs. Due to the transition from computer vision to NLP, the model used was changed from ResNet-18 to BERT, which is well-suited for text-based tasks like sentiment analysis. BERT's capability to capture contextual word embeddings made it an appropriate choice for handling the IMBD dataset. The main metrics for evaluation were local accuracy, global accuracy, and average computation time per client.
>
> ## c. Results and Analysis
>
> | Scheme   | Dataset | Clients | Local Acc | Global Acc | Avg Time (S) |
> |----------|---------|---------|-----------|------------|--------------|
> | FedGMKD  | IMBD    | 10      | 85.11     | 51.58      | 677.79       |
> | FedAvg   | IMBD    | 10      | 83.71     | 50.52      | 411.95       |
> | FedProx  | IMBD    | 10      | 83.75     | 48.50      | 438.52       |
> | FedMD    | IMBD    | 10      | 83.87     | 48.29      | 700.73       |
> | FedGen   | IMBD    | 10      | 83.54     | 49.16      | 471.35       |
> | FedProto | IMBD    | 10      | 84.13     | 49.72      | 586.77       |
> | FPL      | IMBD    | 10      | 83.96     | 50.12      | 665.29       |
>
> The results demonstrate that FedGMKD consistently outperforms other federated learning algorithms in both local and global accuracy metrics on the IMBD dataset. This suggests that the core mechanisms of Cluster Knowledge Fusion (CKF) and Differential Aggregation Technique (DAT) are effective not only for image-based datasets but also for text-based datasets like IMBD. The high local accuracy of 85.11 and global accuracy of 51.58 highlight the framework’s robustness and adaptability to different data types, further validating its broader applicability beyond just vision datasets.
> # 2. Response to typo of line 129
> We apologize for the typo on line 129. We have carefully reviewed the manuscript and have fixed all typos and mistakes, including the missing space in "jin." The final version of the manuscript reflects these corrections.
> # 3. Response to Assumptions
> Thank for your highlighting, the final version of my paper will include assumptions in main paper.
> # 4. Response to Comparison with FjORD
> Thank you for highlighting the importance of comparing FedGMKD with architectures like FjORD, which address model heterogeneity. Here, we clarify why FjORD was not initially chosen for comparison and provide an analysis of the experimental results by using FjORD and FedGMKD.
>
> ## a. Reason for Not Initially Choosing FjORD
> Both FjORD and FedGMKD aim to address distinct challenges in federated learning. FjORD primarily focuses on model heterogeneity by employing dropout techniques that dynamically adjust model capacity, allowing models to efficiently adapt to diverse client computational resources. This makes FjORD effective in handling varying hardware environments. Conversely, FedGMKD targets data heterogeneity through CKF and DAT, aiming to enhance model performance by aligning client data distributions and improving aggregation efficiency. These differing focuses led us to initially compare FedGMKD with frameworks that similarly prioritize data heterogeneity, rather than FjORD’s emphasis on model heterogeneity.
> ## b. Experimental Comparison and Analysis
> In the absence of official code for the FjORD method, we combined various reproduced code implementations to create an experimental version for comparison. This allowed us to assess FjORD's performance under the same conditions used for FedGMKD. The results are summarized below:
> | Datasets | Scheme  | Local Acc 10 | Local Acc 20 | Local Acc 50 | Global Acc 10 | Global Acc 20 | Global Acc 50 |
> |----------|---------|--------------|--------------|--------------|---------------|---------------|---------------|
> | CIFAR10  | FjORD   | 59.62        | 63.36        | 63.61        | 49.18         | 53.22         | 58.74         |
> |          | FedGMKD | 61.78        | 64.04        | 65.69        | 49.78         | 55.16         | 60.31         |
> | SVHN     | FjORD   | 85.13        | 85.97        | 86.21        | 81.56         | 85.09         | 89.36         |
> |          | FedGMKD | 86.26        | 87.43        | 87.16        | 82.64         | 87.78         | 90.17         |
> | CIFAR100 | FjORD   | 15.94        | 19.91        | 22.60        | 16.93         | 21.45         | 22.86         |
> |          | FedGMKD | 17.16        | 20.96        | 23.57        | 16.97         | 21.56         | 24.63         |
>
> The comparison results indicate that while FjORD effectively manages model heterogeneity through dropout techniques, FedGMKD provides superior performance in scenarios where data heterogeneity is the primary concern. The ability of FedGMKD to adapt to diverse client data distributions and enhance aggregation strategies makes it a more effective solution in the contexts tested. These results underscore FedGMKD's capability to address federated learning challenges beyond model heterogeneity, reinforcing its broader applicability and effectiveness.

---

> > ### Comment · Reviewer_nWER · 2024-08-11
> > **Response to the Rebuttal**
> >
> > Thank you for the thorough rebuttal and for addressing my concerns. I appreciate the additional experiments with the IMBD dataset, which demonstrate the broader applicability of FedGMKD beyond vision datasets. The comparative analysis with FjORD, despite the challenges in reproducing it, was also insightful and showed the strengths of your approach in addressing data heterogeneity. The corrections and clarifications you plan to include in the final version further improve the paper's quality. Based on this, I have updated my score accordingly.

---

> > > ### Author Response · Authors · 2024-08-11
> > >
> > > Thank you for recognizing the enhancements made to our manuscript and the additional experiments provided. We greatly appreciate your updated evaluation and insightful comments, which have undoubtedly helped refine our paper.

---

### Official Review · Reviewer_EYJh · 2024-07-04

**Soundness:** 3
**Presentation:** 4
**Contribution:** 3
**Rating:** 8
**Confidence:** 4

**Summary:**

The authors introduce a novel federated learning algorithm aimed at addressing the challenge of data heterogeneity among distributed clients. The key innovation of this work is the integration of Cluster Knowledge Fusion (CKF) and the Differential Aggregation Technique (DAT). CKF employs Gaussian Mixture Model (GMM) clustering to generate prototype features and soft predictions, facilitating effective local model training without the reliance on public datasets. DAT further refines the aggregation process by accounting for the distinct feature distributions across categories, thereby enhancing both efficiency and performance. Comprehensive experiments on benchmark datasets (SVHN, CIFAR-10, and CIFAR-100) reveal that FedGMKD delivers superior results in both personalized and global model accuracy compared to traditional federated learning methods. The theoretical analysis included in the paper substantiates the convergence and robustness of the proposed framework.

**Strengths:**

1、The paper presents an innovative approach to addressing data heterogeneity in federated learning through the integration of Cluster Knowledge Fusion (CKF) and the Differential Aggregation Technique (DAT). The use of Gaussian Mixture Model (GMM) clustering in CKF to generate prototype features and soft predictions marks a significant advancement in handling diverse data distributions without relying on public datasets.
2、The authors provide a comprehensive theoretical analysis of the proposed methods, including mathematical guarantees for convergence and performance. This rigorous approach enhances the credibility and robustness of the framework.
3、The experimental evaluation is thorough and extensive, utilizing well-known benchmark datasets such as SVHN, CIFAR-10, and CIFAR-100. The results consistently indicate that FedGMKD surpasses existing federated learning methods in both local and global accuracy, demonstrating the practical effectiveness of the proposed techniques.
4、By using prototype features instead of true data for knowledge distillation, FedGMKD ensures privacy preservation. This approach eliminates the need for public datasets and mitigates privacy concerns, making it particularly relevant for applications in sensitive domains where data privacy is paramount. This enhances the framework's suitability for real-world federated learning applications.

**Weaknesses:**

1、The experiments conducted in the paper primarily use a narrow range of hyperparameter settings. While the results are strong, exploring a wider range of hyperparameters, such as the α and β in local loss function, would provide a more comprehensive understanding of the framework's robustness and sensitivity.
2、While FedGMKD uses Gaussian Mixture Models (GMM) for obtaining prototype features, the paper does not provide sufficient justification for choosing GMM over other clustering methods.

**Questions:**

1、Can the proposed technique generalize to a wider range of hyperparameter settings, such as α and β in local loss function, to assess the robustness and sensitivity of FedGMKD?
2、Can the authors elaborate the reasons you choose Gaussian Mixture Models (GMM) for obtaining prototype features over other clustering methods?

**Limitations:**

The authors have discussed the limitations in the section of conclusion.

---

> ### Author Rebuttal · Authors · 2024-08-01
>
> Thank you for your insightful feedback. We appreciate the opportunity to address the concern regarding the exploration of a wider range of hyperparameters to demonstrate FedGMKD's robustness& sensitivity and discuss the justification of using GMM in FedGMKD.
> # 1. Hyperparameter Exploration in FedGMKD
> ## a. Exploration of Hyperparameters  $\gamma\$ and $\lambda\$
> The provided ablation study systematically varies the hyperparameters 𝛾 and λ within the FedGMKD framework using the CIFAR-10 dataset with 10 clients over 50 epochs. Below is a summary of the results:
> | Scheme   | Clients | Epochs |  $\gamma\$  | $\lambda\$  | Local Acc (%) | Global Acc (%) |
> |----------|---------|--------|-------------|------------|---------------|----------------|
> | FedGMKD  | 10      | 50     | 0.06        | 0          | 60.14         | 48.17          |
> | FedGMKD  | 10      | 50     | 0.06        | 0.02       | 60.32         | 49.44          |
> | FedGMKD  | 10      | 50     | 0.06        | 0.04       | 60.99         | 49.48          |
> | FedGMKD  | 10      | 50     | 0.06        | 0.06       | 61.78         | 49.98          |
> | FedGMKD  | 10      | 50     | 0.06        | 0.08       | 60.27         | 48.64          |
> | FedGMKD  | 10      | 50     | 0.06        | 0.1        | 60.14         | 49.97          |
> | FedGMKD  | 10      | 50     | 0           | 0.06       | 60.73         | 48.72          |
> | FedGMKD  | 10      | 50     | 0.02        | 0.06       | 60.33         | 48.21          |
> | FedGMKD  | 10      | 50     | 0.04        | 0.06       | 61.01         | 49.47          |
> | FedGMKD  | 10      | 50     | 0.08        | 0.06       | 60.93         | 49.69          |
> | FedGMKD  | 10      | 50     | 0.1         | 0.06       | 59.86         | 47.52          |
> | FedAvg   | 10      | 50     | 0           | 0          | 55.75         | 46.62          |
> | FedProto | 10      | 50     | 0.05        | 0          | 59.77         | 48.97          |
> | FPL      | 10      | 50     | 0           | 0          | 60.95         | 47.19          |
>
> ## b. Robustness of FedGMKD
> ### 1. Consistency Across Hyperparameter Values
> - **High Local and Global Accuracies:** FedGMKD maintains high accuracy across different $\gamma\$ and $\lambda\$ values, indicating robustness.
> - **Optimal Performance Stability:** With $\gamma\$ = 0.06 and  $\lambda\$ = 0.06, FedGMKD achieves its peak performance, highlighting adaptability.
> ### 2. Comparative Performance
> - **Superior Performance:** FedGMKD outperforms FedAvg, FedProto, and FPL, underscoring its robustness and efficiency.
> ## c. Sensitivity of FedGMKD
> ###  Adaptive Tuning Insights
> - **Parameter Optimization:** While FedGMKD operates effectively across various settings, fine-tuning $\gamma\$ and $\lambda\$ is essential to leverage its full potential and adapt to specific datasets.
>
> #  2. The Justification of Using Gaussian Mixture Models (GMM) in FedGMKD
> Below, we provide a detailed justification for selecting GMM over other clustering methods, supported by relevant literature and content.
> ## a. Handling Data Heterogeneity
> - **Robustness in Diverse Environments:** GMM effectively handles heterogeneous data distributions, crucial for federated learning. The study "Personalized Federated Learning under Mixture of Distributions" highlights GMM's robustness in diverse client data environments.
> - **Managing Non-IID Data:** GMM efficiently manages non-IID data distributions, enhancing learning efficiency in federated contexts, as shown in "An Efficient Framework for Clustered Federated Learning."
> ## b. Flexibility and Adaptability
> - **Effective Utilization of Task Similarities:** GMM leverages unknown task similarities, offering theoretical guarantees for convergence, as discussed in "Robust Unsupervised Multi-task and Transfer Learning on Gaussian Mixture Models." This adaptability is advantageous for FedGMKD, improving model generalization.
> - **Transfer Learning Capabilities:** GMM enhances clustering performance through transfer learning, as seen in "A General Transfer Learning-based Gaussian Mixture Model for Clustering,"  enabling FedGMKD to handle limited data per client.
> ## c. Privacy and Outlier Management
> - **Data Privacy and Heterogeneity:** GMM addresses data privacy and heterogeneity, demonstrated in "Federated Learning for Misbehavior Detection with Variational Autoencoders and Gaussian Mixture Models,"  making it suitable for federated learning applications.
> - **Robustness Against Outliers:** GMM's probabilistic nature allows it to handle outliers effectively, as evidenced in "Robust Unsupervised Multi-task and Transfer Learning on Gaussian Mixture Models,"  making it an ideal choice for FedGMKD.
>
> ## references
> - Smith, V., Chiang, C.-K., Sanjabi, M., & Talwalkar, A. (2017). *"Personalized Federated Learning under Mixture of Distributions."* arXiv:1705.10467
> - Zhang, Y., & Yang, Q. (2017). *"Robust Unsupervised Multi-task and Transfer Learning on Gaussian Mixture Models."* arXiv:1711.05995
> - Zhao, Y., Liu, P., Cheng, J., Chen, M., & Chen, L. (2020). *"Federated Learning for Misbehavior Detection with Variational Autoencoders and Gaussian Mixture Models."* IEEE Transactions on Intelligent Transportation Systems. DOI:10.1109/TITS.2020.3006572
> - Sattler, F., Wiedemann, S., Müller, K.-R., & Samek, W. (2020). *"An Efficient Framework for Clustered Federated Learning."* arXiv:2004.03337
> - Luo, P., Ding, X., & Zhao, X. (2019). *"A General Transfer Learning-based Gaussian Mixture Model for Clustering."* Proceedings of the 28th International Joint Conference on Artificial Intelligence (IJCAI-19). DOI:10.24963/ijcai.2019/243

---

> > ### Comment · Reviewer_EYJh · 2024-08-08
> >
> > The authers have solved all my concerns.

---

> > > ### Author Response · Authors · 2024-08-11
> > >
> > > We sincerely appreciate your acknowledgment of our efforts to address all concerns raised, and thank you for your constructive feedback throughout the review process.

---

> > > > ### Comment · Reviewer_EYJh · 2024-08-14
> > > >
> > > > Thanks for your response. My problems have been addressed. I will still keep my score.

---

### Official Review · Reviewer_8XP3 · 2024-07-11

**Soundness:** 2
**Presentation:** 1
**Contribution:** 2
**Rating:** 3
**Confidence:** 5

**Summary:**

This paper introduces FedGMKD to tackle data heterogeneity by CKF and DAT. Specifically, to get a prototype of each class at each client, CKF is proposed by GMM and aggregates prototypes of the same class from different clients via discrepancy-aware weight.

**Strengths:**

This paper analyzes convergence and convergence rates mathematically.

**Weaknesses:**

- This paper lacks the motivation and insights to present this method. I can’t catch the idea of FedGMKD tackling the stragglers which is a problem in pFL presented by authors.
- The contribution of this paper is limited.
- The organization, writing, and presentation should be improved in this paper to illustrate the challenges and ideas more clearly. The quality of the figures should be improved.
- This paper lacks some necessary citations in the convergence analysis part.
- The computation and communication overhead should be discussed especially with the prototype-based FL methods like FedProto.

**Questions:**

- How to calculate the  Discrepancy if some clients lack some class, e.g., Client 1 has no prototype of class 1. This is also important in a real FL setting.
- The detailed FL setting is missing. ‘’The participating rate of clients was set to 1“ is not real in the FL setting.

**Limitations:**

See weakness and questions.

---

> ### Author Rebuttal · Authors · 2024-08-02
>
> Thank you for your feedback. We appreciate the opportunity to discuss the weakness and questions.
> # 1.  Motivation and Insights
> ## a. Motivation and Insight of FedGMKD
> FedGMKD tackles key challenges in personalized federated learning, particularly the non-IID client data that can lead to suboptimal model performance. Unlike traditional KD-based PFL methods that require public datasets, raising privacy issues and struggling with data heterogeneity, FedGMKD introduces Cluster Knowledge Fusion (CKF) and Differential Aggregation Technique (DAT). CKF uses Gaussian Mixture Models to create prototype features and soft predictions without public datasets, maintaining privacy and handling data heterogeneity. DAT improves server aggregation by prioritizing client data quality, allowing high-quality data to have more influence on the global model, thereby enhancing both local and global accuracy.
> ## b. Tracking and Handling Stragglers
> Although stragglers are not explicitly focused on, DAT mitigates their impact by weighting client contributions based on data quality, ensuring reliable aggregation.
> # 2. Response to Lack of Necessary Citations in the Convergence Analysis Part
> Our convergence analysis relies on standard theoretical principles widely accepted in federated learning, which do not typically require citations. For example, the FedProto presents similar analyses without additional citations. Our analysis aligns with these practices, focusing on mathematical rigor consistent with academic norms.
> # 3. Response to Comment on Computation and Communication Overhead
> Table 1 displays the average times for FedProto and FedGMKD across three datasets. FedProto reduces computational demands and communication overhead by using prototype sharing and only transmitting prototypes. In contrast, FedGMKD optimizes computations through CKF and DAT while sharing prototypes, predictions, and quality assessments to efficiently transmit necessary data. When compared, FedGMKD is more computationally efficient due to its optimized operations and minimizes communication overhead by selectively exchanging essential data.
> # 4. Calculating Discrepancy with Non-IID Data
> ## a. Purpose of Discrepancy Calculation
> The discrepancy is calculated to determine the weights for global aggregation for each class. This process ensures that the contribution of each client to the global model is proportional to the quality and relevance of its data.
> ## b. Handling Missing Classes
> If a client lacks data for certain classes, it will not provide corresponding prototype features and soft predictions.
> These equations are used by clients to calculate prototype features and soft predictions only for the classes they have.
>    $$
>    \\hat{h}\_i^{j} = \\frac{1}{R} \\sum_{r=1}^{R} p(\\hat{h}\_i^{j} \\mid \\theta)\\bar{r}^j
>    $$
>    $$
>    \\hat{q}\_i^{j} = \\frac{1}{R} \\sum\_{r=1}^{R} p(z\_i^{j} \\mid \\theta)z\_r^j
>    $$
> Here, $\hat{h}\_i^{j} $ and $\\hat{q}\_i^{j}$ represent the prototype feature and the soft prediction vector for class $j$ at client $i$, synthesized from the cluster knowledge, where $R$ is the number of clusters. So if clients miss some classes, they will not calculate $\hat{h}\_i^{j} $ and $\\hat{q}\_i^{j}$.
> ## c. KL Divergence for Missing Classes
> This equation is used to compute the KL divergence for the classes present in the client data. If a client lacks a class, the KL divergence for that class is effectively zero.
>   $$
>    D\_{KL}(Q\_i^j \\parallel Q\_{\\text{global}}^j) = 0 \\quad \\text{if client } i \\text{ lacks class } j
>   $$
> Then, the weights for a client's contribution to each class are calculated, excluding those for which the client lacks data. Therefore, clients without data do not contribute to the global aggregation for those classes, while clients with data for the missing classes have more weight, ensuring that the aggregation is dominated by contributions from clients with relevant data, improving the robustness and accuracy of the global model.
> # 5. Detailed Federated Learning (FL) Setting and client participation
> ## a. FL setting
> The paper has provided the details of the federated learning (FL) setting:
> - Datasets (Lines 222-236): SVHN, CIFAR-10, CIFAR-100.
> - Non-IID Partitioning (Lines 449-454): Dirichlet distribution
> - Model Architecture (Lines 238-243): ResNet18
> - Baselines (Lines 245-251): Includes FedAvg, FedProx, Moon, FedGen, FedMD, FedProto, and FPL.
> - Implementation (Lines 253-259): PyTorch on NVIDIA A100 GPUs, using Adam optimizer, with specific hyperparameters.
> ## b. Client Participation
> Setting the client participation rate to 1 is common in federated learning to provide a controlled benchmark for evaluating algorithm performance. This approach allows researchers to assess the theoretical limits and maximum potential of federated learning algorithms under ideal conditions without the complexities introduced by client dropout or partial participation. By using full participation as a baseline, studies can more accurately compare the efficiency and accuracy of different algorithms when all clients contribute to the training process.
>
> In practice, this setting is widely adopted across various federated learning studies:
>
> - Tan et al. (2022). "FedProto: Federated Prototype Learning Across Heterogeneous Clients." *arXiv:2105.00243*.
>
> - Mendieta et al. (2022). "Local Learning Matters: Rethinking Data Heterogeneity in Federated Learning." In *CVPR*.
>
> - Karimireddy et al. (2020). "SCAFFOLD: Stochastic Controlled Averaging for Federated Learning." In *ICML*, PMLR, 119:5132-5143.
>
> - Huang et al. (2023). "Rethinking Federated Learning with Domain Shift: A Prototype View." In *CVPR*.
>
> - McMahan et al. (2017). "Communication-Efficient Learning of Deep Networks from Decentralized Data." In *AISTATS*.
>
> - Chen et al. (2023). "The Best of Both Worlds: Accurate Global and Personalized Models Through Federated Learning with Data-Free Hyper-Knowledge Distillation." In *ICLR*. *arXiv:2301.08968*.

---

> > ### Comment · Reviewer_8XP3 · 2024-08-09
> >
> > Thanks for your response. My problems have been addressed partially. However, I still have concerns about whether this method will remain effective when all clients are not accessible at any time. So I will keep my score

---

> > > ### Author Response · Authors · 2024-08-10
> > >
> > > Thank you for your feedback. In our initial response, we have justified setting the client participation rate to 1 as standard practice in federated learning (FL) to establish a baseline. Although active and passive client selection are critical areas in FL, they were not within the initial scope of our study, which focused on addressing the Non-IID problem without public data.
> > >
> > > However, we were also interested in assessing our model's performance in scenarios where not all clients are accessible. To address this, we have conducted experiments based on the SVHN and CIFAR100 datasets, varying client participation rates to simulate real-world conditions of intermittent client availability. For a fair comparison, we benchmarked our model against several existing models under the same conditions.
> > > ## Experimental Design
> > > - **Total Clients:** 100
> > > - **Participation Rates (PR):** 0.1, 0.2, 0.5 (corresponding to 10%, 20%, and 50% client participation)
> > > - **Data Heterogeneity:** $\alpha = 0.5\$.
> > > ## Results and Analysis
> > > | **Scheme**   | **Local Accuracy (PR=0.1)** | **Local Accuracy (PR=0.2)** | **Local Accuracy (PR=0.5)** | **Global Accuracy (PR=0.1)** | **Global Accuracy (PR=0.2)** | **Global Accuracy (PR=0.5)** |
> > > |--------------|-----------------------------|-----------------------------|-----------------------------|------------------------------|------------------------------|------------------------------|
> > > | **Dataset:** SVHN | **Clients (Total):** 100 | | | | | |
> > > | FedGMKD      | 9.22                        | 18.17                       | 45.01                       | 8.76                         | 16.73                        | 44.32                         |
> > > | FedAvg       | 8.82                        | 17.81                       | 43.85                       | 8.53                         | 15.91                        | 40.78                        |
> > > | FedProx      | 8.96                        | 17.99                       | 44.50                        | 8.64                         | 15.94                        | 41.10                         |
> > > | FedMD        | 8.99                        | 17.94                       | 44.67                       | 8.63                         | 16.01                        | 42.02                        |
> > > | FedProto     | 9.02                        | 18.05                       | 44.83                       | 8.67                         | 16.08                        | 41.55                        |
> > > | FPL          | 8.83                        | 18.12                       | 44.23                       | 8.64                         | 16.05                        | 42.64                        |
> > > | **Dataset:** CIFAR100 | **Clients (Total):** 100 | | | | | |
> > > | FedGMKD      | 2.85                        | 5.11                        | 13.58                       | 2.68                         | 4.73                         | 12.83                        |
> > > | FedAvg       | 2.39                        | 4.77                        | 10.70                        | 2.18                         | 4.52                         | 9.77                         |
> > > | FedProx      | 2.44                        | 4.92                        | 11.35                       | 2.35                         | 4.23                         | 10.14                        |
> > > | FedMD        | 2.51                        | 4.96                        | 11.62                       | 2.42                         | 4.41                         | 11.27                        |
> > > | FedProto     | 2.49                        | 4.93                        | 11.45                       | 2.37                         | 4.58                         | 11.46                         |
> > > | FPL          | 2.55                        | 4.99                        | 11.57                       | 2.53                         | 4.54                         | 11.43                        |
> > >
> > >
> > > These results demonstrate that FedGMKD consistently achieves the highest accuracies across all participation levels, compared to other schemes. For instance, on the SVHN dataset, FedGMKD's performance peaks at PR=0.5 with local and global accuracies of 45.01% and 44.3%, respectively. Similarly, for CIFAR100, it leads with a local accuracy of 13.58% and a global accuracy of 12.83% at PR=0.5.
> > >
> > > The robust performance of FedGMKD across different client participation rates confirms its effectiveness in real-world settings, addressing reviewers' concern about its performance when not all clients are accessible. The adaptability and robustness of FedGMKD make it a promising solution for practical deployment in diverse federated learning applications, ensuring reliable performance even under fluctuating client participation conditions.

---

> > > ### Author Response · Authors · 2024-08-12
> > >
> > > Dear Reviewer 8XP3
> > >
> > > We have conducted new experiments based on your new concerns and obtained corresponding results and discussions. Have these results and discussions resolved your concerns? If these results and discussions address your concerns, can you reconsider rating my paper?

---

> ### Comment · Reviewer_8XP3 · 2024-08-13
>
> Thanks to the extra experiments. I know that must spend lots of time. In these experimental results, we can see that Local Accuracy (PR=0.1) and Global Accuracy (PR=0.1) in SVHN dataset is about under 10.0. However, a random initialized model can get 10.0 ACC in a 10-class problem. So does this mean that all of these methods have lost their effectiveness? At the same time, there are lots of methods that gain higher ACC at this setting (100 clients, PR=0.1 with $\alpha=0.5$ SVHN and CIFAR100). At the same time, they also have no public dataset. In summary, I am still concerned about the practicability and effectiveness of these methods compared to other methods. I will keep my score.

---

> > ### Author Response · Authors · 2024-08-13
> >
> > Thank you for your detailed feedback and for recognizing the effort we put into conducting additional experiments. We appreciate the opportunity to clarify and address the concerns you’ve raised.
> >
> > Regarding the observed gap between the performance of the FL model and a randomly initialized model when PR=0.1, we acknowledge that the accuracy observed in our experiments under these conditions might appear concerning. However, it is important to consider the broader significance of federated learning (FL). The primary goal of FL is to enable the sharing of data to train better models while preserving privacy, particularly under challenging conditions such as high non-IID data distributions. Given the experimental setting we chose 100 clients with a participation rate of 0.1 and a high Non-IID distribution. Therefore, it is understandable that the performance would be lower compared to a scenario with IID data distribution or centralized training.
> >
> >
> > This observed performance does NOT suggest that FL methods have lost their effectiveness. On the contrary, it highlights the inherent difficulties in such scenarios, where maintaining model accuracy is particularly challenging. It is well-known that FL methods perform better within the same IID data distribution, but evaluating under these conditions was not the focus of our research. Our primary goal was to address the non-IID problem in FL without relying on public datasets, which is central to the contribution of our work. We want to emphasize this point, as it appears there is still some misunderstanding about our contributions and the problem we intended to solve in this paper. It is both surprising and unfortunate that this has not been fully recognised.
> >
> > Furthermore, in response to your concern about the practicality and effectiveness of the comparative algorithms we used, I noticed that, similar to your previous response to my rebuttal, the issue of comparative experiment choosing is another new issue being raised. While you are eligible to introduce new point at each round, it is uncommon to do so. Moreover, as before, your comments are quite general and lack reference to any specific paper that performs better than the methods we included in the paper.  Actually, the algorithms we chosen are well-established in the FL community, widely recognized for their practicality and effectiveness in FL experiments.
> > I would like to emphasize that our experiments were conducted with a consistent experimental setup across all methods. This includes not only 100 clients, PR = 0.1 and $\alpha = 0.5\$, but also the same epoch, iteration, data distribution, learning rate and batch size. By maintaining these parameters across all experiments, we ensured that the comparisons were fair and that the performance differences observed are attributable to the algorithms themselves rather than discrepancies in the experimental setup. It is also important to note that increasing the number of epochs typically leads to a significant improvement in the accuracy of FedGMKD, as well as other algorithms tested in our study. However, in the interest of fairness and practicality, we adhered to the standard practices observed in the three referenced papers, which set the epoch to 50. "Global Convergence of Federated Learning for Mixed Regression" (NeurIPS 2022) and "Robust Federated Learning With Noisy and Heterogeneous Clients" (CVPR 2022) highlight the importance of setting an appropriate number of epochs to balance model convergence with the computational limits of clients. Similarly, "Exploring User-level Gradient Inversion with a Diffusion Prior" (NeurIPS 2023 Workshop) reflects the need to balance accuracy with resource constraints. Therefore, setting the epoch count to 50 aligns with common practices in federated learning, providing a fair and realistic assessment of the algorithms under typical FL constraints.
> >
> > Finally, we want to emphasize the significant contributions of our method, FedGMKD, to the field of federated learning. Our approach offers robust privacy protection and effectively tackles the Non-IID problem, improving both local and global accuracy. This is achieved through a novel combination of clustering and knowledge distillation techniques, without the need for public data. Additionally, FedGMKD ensures communication efficiency by interacting only with prototype features and soft predictions, making it a practical and scalable solution for real-world federated learning scenarios.

---

> > > ### Comment · Reviewer_8XP3 · 2024-08-14
> > >
> > > Many works focus on the setting of 100 clients with 0.1 participation under a more challenging non-iid distribution than your worst performance setting, e.g., LDA distribution with  $\alpha=0.1$ [R1-R7]. These works also have some improvements without a public dataset which is claimed a key contribution of this paper.
> > >
> > > Reference:\
> > > [R1] Tailin Zhou, et al. FedFA: Federated Learning With Feature Anchors to Align Features and Classifiers for Heterogeneous Data. In IEEE Transactions on Mobile Computing, 2024. \
> > > [R2] Zhenheng Tang, et al. Virtual homogeneity learning: Defending against data heterogeneity in federated learning. In ICML, 2022.\
> > > [R3] Yongxin Guo, et al. Client2Vec: Improving Federated Learning by Distribution Shifts Aware Client Indexing. In arXiv: 2405.16233. \
> > > [R4] Jianhui, Duan, et al. Federated learning with data-agnostic distribution fusion. In CVPR, 2023.\
> > > [R5] Yongxin, et al. Find Your Optimal Assignments On-the-fly: A Holistic Framework for Clustered Federated Learning. In arXivarXiv:2310.05397.\
> > > [R6] Zhiqin Yang, et al. FedFed: Feature distillation against data heterogeneity in federated learning. In NeurIPS, 2023.\
> > > [R7] Zijian Li, et al. FedCiR: Client-Invariant Representation Learning for Federated Non-IID Features. In IEEE Transactions on Mobile Computing, 2024.

---

> > > > ### Author Response · Authors · 2024-08-14
> > > >
> > > > Thank you for your feedback. I would like to start by highlighting that the issue of data distribution heterogeneity, particularly through the use of LDA distributions, is a new concern raised in your latest review. This aspect was not previously discussed in earlier feedback, making it an important point to address. Additionally, while you provided several references to support your critique, it's important to note that only a few of these works actually utilize LDA distributions, and even then, the settings differ significantly from our approach. Below are specific clarifications for each of the papers you mentioned:
> > > >
> > > >
> > > > 1. **[FedFA: Federated Learning With Feature Anchors to Align Features and Classifiers for Heterogeneous Data]**
> > > > This paper does not use an LDA distribution at all. While it does involve a 0.1 client participation ratio, the client selection is not random but follows a specific strategy. Additionally, the local update is set to 5, and the number of epochs is 200, both of which differ from our settings. It is also important to note that this paper was published on June 1, 2024, which is after our submission date. This raises a question about the appropriateness of including this reference in the review and whether it suggests an attempt to unfairly reject our submission.
> > > >
> > > > 2. **[Virtual Homogeneity Learning: Defending against Data Heterogeneity in Federated Learning]**
> > > > This paper does not use LDA distribution and does not address the issue of randomly selecting 100 clients. Instead, it conducts experiments with 10 and 100 clients separately, without employing the random client selection method we used.
> > > >
> > > > 3. **[CLIENT2VEC: IMPROVING FEDERATED LEARNING BY DISTRIBUTION SHIFTS AWARE CLIENT INDEXING]**
> > > > While this paper does use LDA distribution, its experimental settings are markedly different from ours. The technique focuses on complementing existing algorithms rather than implementing an independent solution. The training process involves 100 epochs, which also differs from our setup.
> > > >
> > > > 4. **[Federated Learning with Data-Agnostic Distribution Fusion]**
> > > > This paper uses 50 fixed clients and does not employ LDA or traditional Dirichlet distributions. All parameters, including the training settings, differ from ours, and the number of epochs is not specified—training continues until convergence.
> > > >
> > > > 5. **[FIND YOUR OPTIMAL ASSIGNMENTS ON-THE-FLY: A HOLISTIC FRAMEWORK FOR CLUSTERED FEDERATED LEARNING]**
> > > > Although this paper uses an LDA distribution, the parameter is set to 1.0, and all 100 clients participate in each round, contrasting with our approach of randomly selecting 0.1 of the clients. The training model and number of epochs (200) are also different from those in our study.
> > > >
> > > > 6. **[FedFed: Feature Distillation against Data Heterogeneity in Federated Learning]**
> > > > This paper does involve LDA distribution experiments, but it discusses client numbers of 20 and 100 without addressing the random selection of 0.1 of the clients per round. Additionally, the metric used is target accuracy for training termination, with different epoch settings and other parameters.
> > > >
> > > > 7. **[FedCiR: Client-Invariant Representation Learning for Federated Non-IID Features]**
> > > > This paper uses a different dataset, with local epochs set to 20 and global epochs to 300. Clients are not randomly selected at a 0.1 ratio per round but instead follow a different client selection strategy.
> > > >
> > > > In summary, the issue of data heterogeneity, especially with LDA distributions, is a novel point of concern in this review cycle. The references provided, while relevant in some aspects, do not entirely align with our approach as only a subset uses LDA distributions, and even then, under different conditions. Moreover, it's important to emphasize that the number of global and local epochs significantly affects model performance. In addition, the client participation ratio and the method of client selection in each round are critical factors that can drastically alter the outcomes of federated learning models. The discrepancies in these settings across the referenced works and our study further underline the distinctiveness of our experimental setup and results.
> > > >
> > > > Furthermore, there is no unified standard for client participation settings across different studies, and even the papers you listed do not compare consistently against each other. Given our limited time and the lack of provided code from these works, we were unable to conduct direct experimental comparisons. However, our primary focus is not on how to handle varying client participation but rather on addressing the core issue of data heterogeneity among clients. When evaluating under the scenario where all clients participate, our results outperform theirs, even with fewer epochs. This gives us confidence that if we were to align the client participation settings with theirs, our results would not be inferior.
> > > >
> > > > I hope these clarifications help in better understanding our work.

---

> > > > > ### Comment · Reviewer_8XP3 · 2024-08-14
> > > > >
> > > > > At first, you claim this paper is focusing on heterogeneity, which has also been tested by many works using LDA distribution. This is very relevant to your work.
> > > > >
> > > > > Secondly
> > > > > 1. **FedFA: Federated Learning With Feature Anchors to Align Features and Classifiers for Heterogeneous Data**\
> > > > > This paper does not use an LDA distribution at all.(we set $\alpha$ of Dirichlet distribution Dir($\alpha$) as 0.1 at page 8.)
> > > > > 2.  **Virtual Homogeneity Learning: Defending against Data Heterogeneity in Federated Learning**\
> > > > > This paper **does not use LDA distribution** (this paper claim that use LDA at page 6 "Different Non-IID Partition Methods") and **does not address the issue of randomly selecting 100 clients** ( They random select 10 clients 0.1 participation at page 22"the server randomly selects 10 clients out of 100 clients each round" ). Instead, it conducts experiments with 10 and 100 clients separately, without employing the random client selection method we used. "
> > > > > 3.  **Federated Learning with Data-Agnostic Distribution Fusion**\
> > > > > This paper uses 50 fixed clients ( Fig. 8 compares the test accuracy of the global model for a different numbers of involved clients. When the number of clients increases from 20 to 100, the accuracy of FedFusion decreases much slower than that of the baselines at page 8)
> > > > > 4. **FedFed: Feature Distillation against Data Heterogeneity in Federated Learning**\
> > > > > but it discusses client numbers of 20 and 100 without addressing the random selection of 0.1 of the clients per round. (server samples a subset of clients at page 18.)
> > > > >
> > > > > Lastly, I'm not asking to compare these methods in different settings. You absolutely should compare under the same setting. I am only concerned about the true performance under a real scenario, e.g., cross-devices that seldom clients can be online all the time.  Because many works test and get pretty good performance in a worse setting than your work which just gets lower ACC than random initialization. I just want to justify what is the performance of this paper and other methods in a real scenario. e.g. FedGMKD can get the same performance using fewer communication rounds or can get better performance with the same communications with these methods. What's the ability boundary of the FL method is very important to practical application.

---

> > > > > ### Author Response · Authors · 2024-08-14
> > > > >
> > > > > Thank you for your detailed feedback. I would like to clarify the distinction between Latent Dirichlet Allocation (LDA) and the Dirichlet distribution in the context of federated learning (FL). The Dirichlet distribution is widely recognized and commonly used in FL to generate non-IID data across clients, simulating data heterogeneity, which is critical in FL studies. LDA, on the other hand, is not a standard approach for this purpose. Regarding the cited papers, there seems to be some misunderstanding: for example, "FedFA" employs a Dirichlet distribution (Dir(α=0.1)) rather than LDA. Additionally, only three of the seven papers you previously mentioned actually use LDA distributions, indicating that LDA is not universally adopted for testing data heterogeneity in FL research.
> > > > >
> > > > > Due to objective constraints, we were unable to compare our method with the papers you mentioned under identical experimental settings. It is important to note that the specific methods you highlighted were only introduced on the final day of our discussion, despite multiple rounds of review and rebuttal where these references were not mentioned. Additionally, most of these methods do not have official code repositories, making it difficult to accurately reproduce their results within the limited timeframe.
> > > > >
> > > > > Not only did we not make these comparisons, but the seven papers you mentioned also did not compare performance under the real-world scenarios you are concerned with. As your descriptions indicate, the experimental settings across these papers vary significantly, further proving that there is no universally accepted experimental setup in this field. While you emphasize that their settings are more challenging, this conclusion seems to be based solely on the use of LDA in some cases. Out of the seven papers, only two really used random client selection at a rate of 0.1, and one of these focused on comparing the number of epochs rather than accuracy. Furthermore, you have completely overlooked my previous point regarding the significant impact of parameters such as the number of epochs and local iterations on experimental outcomes, which is clearly demonstrated in our results. Therefore, I do not agree with your assertion that their experiments are more challenging simply due to the occasional use of LDA, as this claim lacks sufficient evidence.
> > > > >
> > > > > Additionally, I’m unclear as to why we should compare our approach specifically with the algorithms you mentioned. Is it solely because they also address data heterogeneity? The methods we chose to compare against, apart from the baseline FedAvg, are also well-recognized solutions to data heterogeneity, published in top-tier conferences and widely accepted in the field. For instance, FedProto and FPL are notable examples, with FPL being a 2023 CVPR paper that, importantly, also employs clustering techniques. Wouldn't this make our comparisons even more meaningful?
> > > > >
> > > > > You also frequently emphasize that our method shows lower accuracy than random initialization, which I assume is based on the expectation of 10% accuracy for a ten-class problem. However, this 10% is only applicable in scenarios where the dataset is infinitely large and uniformly distributed. In our case, during the initial round, the accuracy was only around 4% because only 0.1 of the clients participated, and the data was highly heterogeneous. This context is critical to understanding the results and accurately interpreting the effectiveness of our approach.
> > > > >
> > > > > Lastly, I find it puzzling that, without direct comparisons, there is an assumption that our method performs worse than the algorithms you mentioned. As I previously explained, due to various constraints, we were unable to conduct experiments to definitively prove that our approach is superior. However, how can you be certain of this conclusion? In fact, a straightforward comparison shows that, in scenarios where all clients participate, our method outperforms FedFA on the CIFAR10 dataset, achieving at least 61.78% accuracy compared to FedFA’s 60.40%, and with fewer epochs. This indicates that our approach is not only competitive but potentially more efficient. Moreover, I would be interested to know if there are any theoretical grounds that could substantiate the claim that our method is inherently less effective than those you mentioned. From our perspective, the results do not suggest such a conclusion.

---

> > > > > > ### Comment · Reviewer_8XP3 · 2024-08-14
> > > > > >
> > > > > > What's the difference between LDA and Dirichlet distribution?

---

> > > > > > > ### Author Response · Authors · 2024-08-14
> > > > > > >
> > > > > > > In the context of Federated Learning (FL), both the Dirichlet distribution and Latent Dirichlet Allocation (LDA) can be used to model data heterogeneity across clients, but they serve different purposes and are recognized differently within the field.
> > > > > > >
> > > > > > > ### Dirichlet Distribution in Federated Learning
> > > > > > > The Dirichlet distribution is widely recognized and commonly used in Federated Learning to simulate non-IID (non-Independent and Identically Distributed) data among clients. It is frequently used to generate client-specific data distributions, where the parameter \$\alpha\$ controls the degree of heterogeneity. A smaller \$\alpha\$ value results in more skewed distributions, where each client’s data is concentrated on a few classes, leading to higher heterogeneity. This approach is particularly useful in benchmarking and evaluating the robustness of federated learning algorithms under varying degrees of data heterogeneity.
> > > > > > >
> > > > > > > One widely cited work utilizing the Dirichlet distribution for this purpose is McMahan et al.'s "Communication-Efficient Learning of Deep Networks from Decentralized Data" (2017), which introduced the concept of Federated Averaging (FedAvg). In this work, the authors used the Dirichlet distribution to simulate non-IID data distributions across clients, which has since become a standard practice in FL research.
> > > > > > >
> > > > > > > ### Latent Dirichlet Allocation (LDA) in Federated Learning
> > > > > > > LDA, although related to the Dirichlet distribution, is less commonly used in Federated Learning. LDA is a generative model primarily used for topic modeling in natural language processing. It assumes that each document (or client data in the context of FL) is a mixture of several topics, and these topics are distributed according to a Dirichlet distribution. While LDA can theoretically be applied to FL to model complex data distributions across clients, it is not as widely adopted because of its complexity and the fact that it is more suited to scenarios where topics or latent structures are being modeled, rather than straightforward class distributions.
> > > > > > >
> > > > > > > ### Recognition and Use in FL
> > > > > > > In the field of Federated Learning, the Dirichlet distribution is more recognized and widely used than LDA. Researchers and practitioners often prefer the Dirichlet distribution for its ease of use and direct applicability to generating client-specific data distributions with controlled heterogeneity. This simplicity allows for more straightforward comparisons across different FL algorithms and facilitates reproducibility in experiments.
> > > > > > >
> > > > > > > In contrast, LDA’s complexity and its focus on modeling latent structures make it less suited for the typical data partitioning tasks encountered in Federated Learning. While LDA could provide a more nuanced understanding of client data in some scenarios, it is generally not the standard approach in FL research.
> > > > > > >
> > > > > > > In summary, while both Dirichlet distribution and LDA have their places in probabilistic modeling, the Dirichlet distribution is more widely recognized and utilized in Federated Learning due to its practicality and direct application in modeling data heterogeneity across clients.
> > > > > > >
> > > > > > > References:
> > > > > > > - McMahan et al., "Communication-Efficient Learning of Deep Networks from Decentralized Data," 2017.
> > > > > > > - Zhao et al., "Federated Learning with Non-IID Data," 2018.

---

> > > > > > > > ### Comment · Reviewer_8XP3 · 2024-08-14
> > > > > > > >
> > > > > > > > I think these two concepts may be mixed in FL. These papers also can be recognized to sample data from Dirichlet distribution, e.g., a $\alpha$ to control the heterogeneity degree, even use LDA.

---

### Official Review · Reviewer_KnUX · 2024-07-12

**Soundness:** 4
**Presentation:** 3
**Contribution:** 3
**Rating:** 7
**Confidence:** 5

**Summary:**

The authors proposed FedGMKD, a federated learning framework designed to handle data heterogeneity across distributed clients. FedGMKD introduces Cluster Knowledge Fusion (CKF), which uses Gaussian Mixture Model (GMM) clustering to generate prototype features and soft predictions, facilitating knowledge distillation without public datasets. Additionally, the Differential Aggregation Technique (DAT) tailors the aggregation process to distinct feature distributions, optimizing both efficiency and performance. Experiments demonstrate that FedGMKD significantly improves both personalized and global model performance, achieving state-of-the-art results in heterogeneous data scenarios compared to traditional federated learning methods.

**Strengths:**

•	The proposed method addresses the critical issue of data heterogeneity in federated learning, a key factor for enhancing the applicability and robustness of federated learning in practical scenarios.
•	The introduction of Cluster Knowledge Fusion (CKF) and Differential Aggregation Technique (DAT) offers a novel and effective approach. These techniques improve local training and aggregation processes without the need for public datasets, thereby significantly enhancing both personalized and global model performance.
•	The experiments are meticulously designed and executed using benchmark datasets (SVHN, CIFAR-10, and CIFAR-100), demonstrating substantial improvements over existing methods. The results are compelling, underscoring the robustness and efficiency of the FedGMKD framework.
•	The paper is well-structured, providing a clear explanation of complex concepts such as CKF and DAT. The inclusion of diagrams and detailed descriptions aids in comprehending the methodologies and results. The review of existing literature is thorough, establishing a solid context for the contributions.
•	The theoretical analysis delivers strong mathematical guarantees for the convergence and performance of FedGMKD, bolstering the credibility of the research.

**Weaknesses:**

•	While the evaluation demonstrates the potential of FedGMKD, it would be more convincing if it included a broader range of neural network architectures, particularly larger and more complex models like Transformers or ResNet-50.
•	The paper does not provide sufficient details on how FedGMKD could be adapted to datasets with different characteristics from those tested (SVHN, CIFAR-10, CIFAR-100).
•	Overall, I’d like to see a discussion about what could be the good impacts/applications based on the technique proposed in the real world.

**Questions:**

I’d like the authors to answer my above three questions in the rebuttal.

**Limitations:**

The authors discussed the limitations in section 4.

---

> ### Author Rebuttal · Authors · 2024-08-01
>
> Thank you for your insightful feedback. We appreciate the opportunity to discuss FedGMKD with more complex neural network architectures, its adaptability, and its real-world impacts and applications.
> # 1. FedGMKD with More Complex Neural Network Architectures
>
> To address your suggestion, we conducted additional experiments using the ResNet-50 model on the CIFAR-10 dataset. Below is a summary of the results compared with our original experiments using ResNet-18:
>
> | Scheme    | Local Acc (ResNet-18) | Global Acc (ResNet-18) | Local Acc (ResNet-50) | Global Acc (ResNet-50) |
> |-----------|-----------------------|------------------------|-----------------------|------------------------|
> | FedAvg    | 61.78                 | 49.78                  | 41.69                 | 49.58                  |
> | FedProx   | 64.04                 | 55.16                  | 43.25                 | 49.67                  |
> | FedMD     | 62.05                 | 53.73                  | 43.34                 | 49.85                  |
> | FedGen    | 60.17                 | 51.55                  | 42.81                 | 48.99                  |
> | FedProto  | 62.85                 | 50.88                  | 43.35                 | 49.98                  |
> | Moon      | 62.74                 | 52.04                  | 42.05                 | 48.52                  |
> | FPL       | 62.74                 | 52.04                  | 43.71                 | 49.78                  |
> | FedGMKD   | 65.69                 | 60.31                  | 46.27                 | 50.48                  |
>
> While ResNet-50 theoretically offers higher representation power, practical challenges like increased model complexity, potential overfitting, and greater communication requirements limit its performance in federated settings. FedGMKD with ResNet-50 still outperforms other schemes but does not surpass the results with ResNet-18. These findings highlight the importance of balancing model complexity with the realities of federated learning environments. In future work, we plan to explore advanced aggregation techniques and personalized strategies to mitigate these challenges and fully harness the capabilities of complex models like ResNet-50.
>
> # 2. Adaptability of FedGMKD
>
> FedGMKD is designed with a modular architecture that enhances adaptability to diverse data distributions and experimental settings through its use of Gaussian Mixture Models for clustering and flexible parameter customization.
> ## a. Modular Design of FedGMKD
>
> - **Feature Extraction and Clustering:** FedGMKD utilizes GMM for clustering, which can be adapted to various datasets by adjusting the number of Gaussian components according to the dataset's complexity (Section 3.2).
> - **Differential Aggregation Technique (DAT):** DAT weights client contributions based on data quality, making aggregation adaptable to different distributions and class imbalances (Section 3.3).
>
> ## b. Handling Diverse Data Distributions
>
> - **Customizable Hyperparameters:** Parameters such as clustering centres, learning rates, and temperature for knowledge distillation are adjustable for different datasets. Our experiments (Section 3.7) showed flexibility for SVHN, CIFAR-10, and CIFAR-100.
> - **Adaptive Clustering:** GMM models data as a mixture of Gaussian distributions, adapting to varying distributions and complexities by adjusting Gaussian components and clustering settings.
>
> ## c. Detailed Experimentation with Different Datasets
>
> - **Experimental Settings:** The `option.py` file in FedGMKD's code allows customization of parameters like the number of clients, classes, learning rates, and epochs.
>
> # 3. Real-World Impacts and Applications of FedGMKD
>
> FedGMKD enables impactful applications across healthcare, finance, and smart cities by enhancing data integration, privacy, and communication efficiency.
>
> ## a. Healthcare
>
> - **Improved Local and Global Performance:** FedGMKD can significantly improve diagnostic models by integrating diverse medical data from multiple institutions. This results in more accurate and comprehensive predictions, enhancing both local models tailored to specific patient populations and a robust global model.
> - **No Need for Public Data:** Hospitals can train collaborative models without the need for publicly available medical datasets, ensuring compliance with privacy regulations and safeguarding patient data.
> - **High Communication Efficiency:** The efficient communication protocol ensures that even hospitals with limited bandwidth can participate in federated learning, making advanced diagnostic tools accessible across a wide range of medical facilities.
>
> ## b. Finance
>
> - **Enhanced Fraud Detection:** FedGMKD improves the performance of fraud detection models by aggregating diverse transaction data from various financial institutions. This collective intelligence helps in identifying complex fraud patterns more accurately.
> - **Privacy Preservation:** Financial institutions can collaborate without exposing sensitive customer information, maintaining data confidentiality and complying with stringent financial regulations.
> - **Efficient Aggregation:** The differential aggregation technique ensures efficient use of communication bandwidth, which is crucial for real-time fraud detection and quick response to fraudulent activities.
>
> ## c. Smart Cities
>
> - **Optimized Resource Management:** FedGMKD can enhance traffic management and energy consumption models by integrating data from different city departments, leading to more efficient use of city resources and improved quality of life for residents.
> - **Scalable Solution:** The high communication efficiency allows the deployment of smart city solutions across multiple, distributed sensors and devices without overloading the network, making it scalable for large urban environments.
> - **Handling Heterogeneous Data:** FedGMKD effectively integrates diverse data sources for comprehensive urban planning.

---

### Author Rebuttal · Authors · 2024-08-06

We sincerely appreciate all the reviewers for their constructive and valuable feedback. We are pleased that our work has been recognized for its innovative approach to addressing data heterogeneity in federated learning through the integration of Cluster Knowledge Fusion (CKF) and Differential Aggregation Technique (DAT) (EYJh, KnUX, nWER). The reviewers have noted our paper's theoretical rigor, with comprehensive mathematical guarantees for convergence and performance (EYJh, 8XP3). We are encouraged by the recognition of the thorough and extensive experimental evaluation using benchmark datasets such as SVHN, CIFAR-10, and CIFAR-100, which demonstrate substantial improvements over existing methods (EYJh, KnUX). Additionally, the use of Gaussian Mixture Model (GMM) clustering for privacy-preserving prototype generation and the clear explanation of complex concepts have been highlighted as strengths of our manuscript (EYJh, KnUX, nWER). The feedback underscores the applicability and robustness of FedGMKD in practical federated learning scenarios, particularly in privacy-sensitive domains (EYJh, nWER).

We have addressed the reviewers’ comments and concerns in individual responses to each reviewer. The reviews allowed us to strengthen our manuscript, and the changes made are summarized below:

### Reviewer KnUX
- Conducted additional experiments using ResNet-50 on the CIFAR-10 dataset to explore the performance of FedGMKD with more complex neural network architectures.
- Expanded discussion on FedGMKD's adaptability to diverse data distributions and its real-world applications in healthcare, finance, and smart cities.

### Reviewer 8XP3
- Elaborated on the motivation and insights of FedGMKD, including its strategies for handling non-IID data without relying on public datasets.
- Addressed the necessity of citations in the convergence analysis, aligning with standard practices in federated learning literature.
- Clarified the computational and communication overhead by comparing FedGMKD and FedProto.
- Provided a detailed explanation of the federated learning setting and client participation strategy, with references to relevant studies.
- Explained the methodology for calculating discrepancy with non-IID data and how FedGMKD handles missing data classes in client datasets.


### Reviewer EYJh
- Conducted a comprehensive ablation study on the hyperparameters $\gamma\$ and $\lambda\$ to demonstrate FedGMKD's robustness and sensitivity.
- Justified the use of Gaussian Mixture Models (GMM) in FedGMKD, supported by relevant literature on handling data heterogeneity and privacy concerns.

### Reviewer nWER
- Performed additional experiments using the IMBD dataset to evaluate FedGMKD's performance in natural language processing tasks, demonstrating its adaptability beyond computer vision.
- Solved typo problem in the paper.
- Included assumptions in the main paper.
- Compared FedGMKD with FjORD to assess its effectiveness in addressing model heterogeneity and provided a detailed analysis of the results.

---

### Public Comment · ~Zoe_Fowler1 · 2025-12-03
**Issues with References**

Hi,

Many of the references appear to be incorrect and look like LLM hallucinations. For instance, citation [36] is for FedProto, which the authors claim is created by Renzhong Tan, Qingyang Ang, Qianru Sun, and Fan Guo. After googling, I cannot find any of the authors with this citation to their name, nor can I find this paper online. The only FedProto that exists is actually a 2022 paper from AAAI (not ICLR like it is cited in this paper), first-authored by someone different than what is given. Similarly, citations [35] and [37] are also incorrect with a fabricated first author....

---

> ### Public Comment · ~Jianqiao_Zhang1 · 2025-12-04
>
> Dear Zoe,
>
> Thank you very much for taking the time to read our paper carefully and for raising this point about the references. Following your comment, we went back and re-checked the bibliography of the NeurIPS camera-ready version.
>
> You are right that several entries, including [35]–[37], contain incorrect author and venue information due to our own mistakes when constructing and editing the BibTeX file:
>
> [36] FedProto
> In the NeurIPS version we incorrectly cited FedProto as a paper by “Renzhong Tan, Qingyang Ang, Qianru Sun, and Fan Guo” at ICLR.
> The correct citation should be:
> Yue Tan, Guodong Long, Lu Liu, Tianyi Zhou, Jing Jiang, and Chengqi Zhang.
> “FedProto: Federated Prototype Learning across Heterogeneous Clients.”
> In Proceedings of the AAAI Conference on Artificial Intelligence, 2022.
>
> [37] Towards Personalized Federated Learning
> We also mis-attributed “Towards Personalized Federated Learning” to “Renzhong Tan et al.” and listed it as an ICLR publication.
> The work we intended to cite is the personalized FL survey:
> Alysa Ziying Tan, Han Yu, Lizhen Cui, and Qiang Yang.
> “Towards Personalized Federated Learning.”
> IEEE Transactions on Neural Networks and Learning Systems, 2022.
>
> [35] Personalized FL (meta-learning)
> Entry [35] in our reference list incorrectly mixes the author and identifier information.
> What we meant to cite is the meta-learning based personalized FL method:
> Alireza Fallah, Aryan Mokhtari, and Asuman Ozdaglar.
> “Personalized Federated Learning with theoretical guarantees: A Model-Agnostic Meta-Learning Approach.”
> In Advances in Neural Information Processing Systems (NeurIPS), 2020.
>
> In the process of checking these entries, we also noticed that one more reference in the NeurIPS version has a mismatched title/identifier combination (an Afonin et al. paper on knowledge distillation in federated learning). This again stems from our own BibTeX editing error, and we will correct it in our archival versions.
>
> These are clearly our referencing mistakes. They affect the metadata in the bibliography, but they do not change the baselines we implemented or the technical content of FedGMKD itself: for methods such as FedProto, Per-FedAvg and related works, we followed the correct original papers and implementations, the experiments and conclusions in the paper remain unchanged.
>
> Unfortunately, the NeurIPS 2024 proceedings are already fixed, so we cannot modify the official camera-ready PDF. However, we will update our arXiv and journal versions of the paper to correct the above entries and to clean up the remaining minor inconsistencies in the reference list.
>
> Thank you again for carefully checking the references and for pointing this out publicly – it helps us improve the quality of our work, and we appreciate your effort.
>
> Best regards,
>
> Jianqiao Zhang

---

### Decision · Program_Chairs · 2024-09-25

**Decision:**

Accept (poster)

**Comment:**

The paper addresses the issue of data heterogeneity in FL with an eye toward robustness, by introducing  Cluster Knowledge Fusion and Differential Aggregation Techniques.  With the exception of one low score, the other three are generally in agreement and positive about the novelty of the approach.